

# Hydrothermal inputs drive dynamic shifts in microbial communities in Lake Magadi, Kenya Rift Valley

Evan R. Collins[1], Troy M. Ferland[2], Isla S. Castañeda[3], R. Bernhart Owen[4], Tim K. Lowenstein[5], Andrew S. Cohen[6], Robin W. Renaut[7], Molly D. O'Beirne[1], Josef P. Werne[1]

[1]Department of Geology and Environmental Science, University of Pittsburgh, Pittsburgh, PA, 15213, United States
[2]Lamont-Doherty Earth Observatory, Columbia University, Palisades, NY, 10964, United States
[3]Department of Earth, Geographic and Climate Sciences, University of Massachusetts Amherst, Amherst, MA, 01003, United States
[4]Department of Geography and LEWI, Hong Kong Baptist University, Kowloon Tong, Hong Kong
[5]Department of Geological Sciences, State University of New York, Binghamton, NY 13902, United States
[6]Department of Geosciences, The University of Arizona, Tucson, AZ 85721, United States
[7]Department of Geological Sciences, University of Saskatchewan, Saskatoon, SK S7N 5E2, Canada

*Correspondence to*: Evan R. Collins (erc92@pitt.edu/ecollins452@gmail.com)

**Abstract.** The Methane Index (MI) is an organic geochemical index that uses isoprenoid glycerol dialkyl glycerol tetraethers (GDGTs) as a proxy for methane cycling. Here, we report results from core spanning > 700 ka in Lake Magadi, Kenya, which shows abrupt shifts between high and low MI values in the core. These shifts coincide with interbedded tuffaceous silt. Where tuffaceous silts are present, MI "switches off" (MI < 0.2); in contrast, where these silts are absent in the core, the MI increases (MI > 0.5). Bulk organic matter is enriched in $^{13}C$ in Magadi during "MI-off" periods, with values of ~ -18‰ in the upper part of the core and -22 to -25‰ in the lower portion. Evidence from *n*-alkanes and fatty acid methyl esters (FAMEs) support previous interpretations of an arid environment with a shallower lake where Thermoproteotal (formerly Crenarchaeota) archaea thrive in a hot spring rich environment over Euryarchaeota. Sediments deposited when the MI switches "on" showed $\delta^{13}C_{OM}$ values as low as -89.4 ‰, but most were within the range of -28 to -30‰, which is consistent with contributions from methanogens rather than methanotrophs. Thus, the likely source of these high MI values in Lake Magadi is methanogenic archaea. Our results show that hydrothermal inputs of bicarbonate-rich waters into Lake Magadi cause a shift in the dominant archaeal communities, alternating between two stable states.



# 1 Introduction

Life thrives in East African soda lakes (Schagerl, 2016) and has been the subject of modern studies of both prokaryotic and eukaryotic organisms, but few have studied their sediments over geologic timescales . Soda lakes represent ~ 18,500 km$^2$ in East Africa (calculated from values in Melack and MacIntyre, 2016). When compared to the three largest African freshwater lakes (lakes Victoria, Tanganyika, and Malawi), these soda lakes account for ~ 13% of the total lake-surface area in East Africa. A survey of microbial isolate diversity in East African lakes found evidence for cyanobacterial and archaeal primary producers with both oxygenic and anoxygenic phototrophs among the microbial population (Grant and Jones, 2016). Unique aerobic and anaerobic heterotrophs that use a variety of electron donors, including sulfur, sulfate, nitrite, carbon dioxide, and methane, were also identified (Grant and Jones, 2016, and sources therein). Near hot-spring outflows, many thermophilic archaea and bacteria isolates were also classified (Grant and Jones, 2016).

Saline alkaline (soda) lakes in the East African Rift Valley often become stratified meromictic bodies of water with a dense monimolimnion below a chemocline (Melack and MacIntyre, 2016). Oxygen rarely penetrates the monimolimnion waters, and as a result, anaerobic bacteria and archaea dominate the bottom waters and sediments. Remineralizing organic matter from the upper water column (mixolimnion) feeds microbial generation of anaerobic oxidation of methane (AOM). Methane-oxidizing microbes, specifically archaeal anaerobic methane-oxidizers (ANME), are coupled to sulfate-reducing bacteria in a microbial consortium (Boetius et al., 2000; Hinrichs and Boetius, 2002; Werne et al., 2004). ANME mediate CH$_4$ levels in freshwater and soda lakes and in modern oceanic systems, and account for approximately 90% of methane consumed through AOM (Egger et al., 2018). Rates of methane consumption differ by environment and type of ANME, with global freshwater systems ranging from 1 to 1 x 10$^5$ nmol CH$_4$ L$^{-1}$ day$^{-1}$ consumed (Martinez-Cruz et al., 2018). Although soda lakes have been less studied, consumption rates as high as 1.6 x 10$^4$ nmol CH$_4$ L$^{-1}$ day$^{-1}$ have been observed in freshwater Lake Kivu (Roland et al., 2018). Tracking AOM over geologic time periods is important because methane release from tropical wetlands was concomitant with the end of glacial conditions in Europe and is poorly



constrained (DeMenocal et al., 2000; Riddell-Young et al., 2023). Additionally, large methane releases might have been  partly responsible for the Permo-Triassic mass-extinction event (Berner, 2002).

Over geologic time, it is possible to gauge periods of increased methane oxidation, as shown by Zhang et al. (2011) in oceanic systems by using a ratio of archaeal GDGT lipids (de Rosa et al., 1977; Langworthy,
1977). The ratio, as described by Zhang et al. (2011), is known as the methane index (MI), which uses GDGTs produced predominantly by Euryarchaeal ANME. The MI has been used to discern methanotrophy using the assumption that benthic methanotrophic Euryarchaeota preferentially produce GDGT-1, -2, and -3, and that GDGTs crenarchaeol (cren) and crenarchaeol' (cren') are thought to come from Thaumarchaeota and Crenarchaeota, which are part of the TACK superphylum, typically found in
the upper water column (Sinninghe Damsté et al., 2002; Pitcher et al., 2009; Zhang et al., 2011).

Currently, the newly suggested names in the Genome Taxonomy Database for Thaumarchaeota and Crenarchaeota are Nitrososphaerota and Thermoproteota, respectively (Oren and Garrity, 2021; Rinke et al., 2021), which are used in this paper. Moreover, Kim and Zhang (2023) have shown a qualitative and
quantitative relationship between the MI and methanotrophy in deep time, namely from the late Oligocene to the early Miocene. Kim and Zhang (2023) showed that the MI is applicable to AOM, with other biomarkers co-occurring in high-MI intervals representative of not only the Group I consortium of anaerobic methanotrophs (ANME) that produce GDGTs, but also of ANME-2 and ANME-3. Until now, no studies have directly applied the MI to sediments in African soda lakes despite evidence for AOM in
modern soda lakes. Combined with MI values, other methane-related indices are used here to interpret methanogenesis and methanotrophy related to AOM. Previous studies have used GDGT-0 and GDGT-2 ratioed to the GDGT crenarchaeol value, which was originally thought to only be produced by mesophilic Thermoproteota. However, the optimum temperature is closer to 40-45 °C (Zhang et al., 2006). Blaga et al. (2009) found that methanogens predominantly produced GDGT-0, whereas Weijers et al. (2011)
showed that methanotrophic archaea predominantly produce GDGT-2.



Lake Magadi (Kenya) is a sulfate-limited lake, and therefore, methanogenesis and methanotrophy may co-occur without suppression of the higher energy yield of sulfate reduction (Nijaguna, 2006; Sorokin et al., 2007; Deocampo and Renaut, 2016; Lameck et al., 2023). Here, we document evidence of methane cycling in Lake Magadi using archaeal isoprenoid glycerol dialkyl glycerol tetraether (GDGT) lipid biomarkers. This study leverages four methane-related indices: (1) the MI; (2) the % GDGT-0/crenarchaeol; (3) % GDGT-2/crenarchaeol; and (4) the ratio of isoprenoid GDGTs ([2] / [3] (hereafter, [2] / [3]) to understand methane cycling in recent and ancient lacustrine sediments. Two distinct communities were found using a combination of the MI and ratios of GDGT-0 and GDGT-2, normalized to crenarchaeol. Intervals of high methanotrophy, as evidenced by MI and % GDGT-2/crenarchaeol, were related to an equally high proportion of methanogens, while in periods when crenarchaeol was dominant, the methane indices were low. Environmental influences on microbial community composition included precipitation/evaporation fluctuations and variations in hydrothermal activity, the latter often related to contemporary tectonics. Leaf wax data from *n*-alkanes was also used to understand changes in the surrounding environment at Lake Magadi. The average chain length of n-alkanes ($ACL_{(25-33)}$) and the carbon preference index of both alkanes and fatty acid methyl esters ($CPI_{alk}$, and $CPI_{FA}$) indicated proportionally higher input of C4 vegetation in an arid landscape. Methane indices were typically higher during periods of reduced hydrothermal activity, indicating more Euryarchaeal communities, whereas Thermoproteota communities thrived during periods of higher hydrothermal activity.

## 2 Materials and methods

### 2.1 Study locations and sampling

Modern Lake Magadi is a seasonally flooded, saline alkaline pan composed of bedded trona ($[Na_3(CO_3)(HCO_3) \cdot 2H_2O]$) located in the southern Kenya Rift near the border with Tanzania (Baker, 1958; Eugster, 1980). Its elevation is approximately 600 to 605 m above sea level (asl) and it has a maximum depth during the rainy seasons season of a few decimeters up to ~ 1 m (Fig. 1; Renaut and Owen, 2023).



The modern lake is fed by ephemeral streams and alkaline hot springs (up to 86°C), distributed along faults around the shoreline (Baker, 1958; Crane, 1981; Allen et al., 1989; Renaut and Owen, 2023). Former high-level shorelines are preserved as coarse clastic sediments and stromatolites around the lake and, with lake sediments of different ages, record deeper paleolakes that occupied the basin during the Quaternary. For example, Casanova (1986, 1987) and Casanova and Hillaire-Marcel reported a paleoshoreline at 956 m asl when the High Magadi Beds (Late Pleistocene-Early Holocene) were being deposited, which implies contemporary water depths of > 40 m. However, these estimates are approximations due to localized uplift and erosion, introducing uncertainty (Owen et al., 2019; Renaut and Owen, 2023). Higher levels have been suggested based on stromatolites but are uncertain (Casanova and Hillaire-Marcel, 1987).Although Lake Magadi is situated near the equator, it, lies in a rain shadow and consequently has a very large moisture deficit (2400 mm evaporation versus 500 mm precipitation annually; Damnati and Taieb, 1995).

Lake Magadi was cored as part of the *Hominin Sites and Paleolakes Drilling Project* (HSPDP) in June 2014 with the aim of furthering our understanding of the paleoenvironments in the East African Rift Valley to better contextualize hominin remains and artifacts, and to understand possible environmental influences on hominin evolution and migration (Cohen et al., 2016). A 197.4-meter core (MAG-14-2A) was drilled in the northern partend of Lake Magadi in June 2014 (1°51'5.76" S; 36°16'45.84" E; Owen et al., 2019). Approximately 107.7 m of sediments were recovered, with a total core recovery of 55.4% (Cohen et al., 2016). Here, we use the age model from Owen et al. (2019). The core ranges from the modern evaporite surface (0 ka) to the Magadi Trachyte basement, dated at ~ 1 Ma at the core site (Owen et al., 2019). Cores were sampled in 2016 during the initial core description at the Continental Scientific Drilling Facility (CSD, formerly LacCore) at the University of Minnesota, Minneapolis. Altogether, 61 samples, covering the period from 456 ka to 14.9 ka (Table 1), were subsampled and freeze-dried from dark brown to black silty clay intervals in the core. These samples were expected to have a high total organic carbon that would yield the best results from organic geochemical studies.



Over the past million years, Lake Magadi has varied from a large subhumid lake, when the freshwater lake was fed by rivers and groundwater continuously, to the small, tectonically restricted, saline alkaline pan partly fed by hot springs (Owen et al., 2019; Renaut and Owen, 2023). From 545 to 380 ka Magadi transitioned to a more pronounced arid system marked by abundant organic-rich sediments (Owen et al., 2019). Periodic freshwater inundation occurred from 380 to 105 ka as marked by interspersed calcite and magnesium-rich calcite at the lake margins, evidence of zeolites formed by trachytic volcanic glass interacting with alkaline water, and sulfate-rich bottom water brines that were subjected to microbial sulfate reduction (Owen et al., 2019; Deocampo et al., 2022). The most recent phases of the lake (105 to 0 ka) were more evaporatively enriched, with abundant trona and minor nahcolite as well as metals (i.e., Br, Pb, Zn, Sb, Ag), indicating increased periods of hydrothermal inflow, especially from ca. 20 to 10 ka (Owen et al., 2019).

## 2.2 Leaf wax and bulk organic preparation and analysis

### 2.2.1 Lipid extraction

To obtain a total lipid extract (TLE), 61 samples from Lake Magadi were freeze dried and homogenized and ca. 5–10 g of sediment were ultrasonically extracted with 2:1 DCM:MeOH. The TLE for each sample was treated with activated copper shot to remove elemental sulfur. The TLEs were then separated into three fractions (apolar (AP), polar one (P1), and polar two (P2)) using activated alumina via a short column chromatography. The AP fraction was eluted with 4 mL of 9:1 Hexane (Hex):DCM (v/v), the P1 fraction with 4 ml 1:1 DCM:MeOH, and the P2 fraction with 4 ml MeOH. The P1 fractions were dried down and re-dissolved in 99:1 Hex:Isopropanol (IPA) (v/v) and filtered through a 0.45 μm 4 mm diameter PTFE filter prior to GDGT analysis.

### 2.2.2 Leaf wax preparation and analysis

Leaf waxes (i.e., free fatty acids and *n*-alkanes) were identified and quantified in the same intervals as GDGTs in the Lake Magadi core. The TLE from the ultrasonic extraction contained free fatty acids (FFAs), which were methylated using $BF_3$ in a methanol solution converting the FFAs to fatty acid methyl



esters (FAMEs). The apolar *n*-alkane fraction did not require methylation to be run. Compound concentrations were determined via gas chromatography-mass spectrometry (GC-MS) using a Thermo Scientific Trace 1310 GC, equipped with an Agilent DB-5 column (30m x 0.320 mm, 0.25 μm film) in tandem with a FID and a Thermo Scientific ISQ QD Single Quadrupole Mass Spectrometer. Samples

were run separately to avoid loss from the FID. The inlet was operated in splitless mode for the FAMEs at a temperature of 250 °C. Column flow was set to 1.5 ml min$^{-1}$ with an initial oven temperature of 70 °C, which was held for 1 minute and then ramped to 130 °C over 6 min at a rate of 10 °C min$^{-1}$, and then ramped to 320 °C over 57.5 min at a step of 4 °C min$^{-1}$ and held at 320 °C for 3 min. The column carrier gas was He and the gas used in the FID was a mixture of air (350 ml min$^{-1}$), He (20 ml min$^{-1}$), and H$_2$ (35

170 ml min$^{-1}$). Quantification of compounds was completed using 5α-androstane as an internal standard and compounds were identified on the FID by comparing to relative retention times with a FAME standard.

### 2.2.4 Bulk organic δ$^{13}$C$_{OM}$ analysis

Samples were subsampled from the same intervals as organic biomarkers for bulk organic carbon isotope analysis. Powdered sediment samples were weighed out in silver capsules and carbonates were removed

by adding 5% HCl in four-hour increments. Samples were analyzed on a Costech Elemental Analyzer coupled to a ThermoFinnigan Delta V Plus isotope ratio monitoring mass spectrometer (IRMS). Samples are reported as per mil (‰) deviations from the Vienna Pee Dee Belemnite (VPDB) standard in conventional delta notation.

### 2.3 GDGT preparation and analysis

### 2.3.1 GDGT analysis

Polar samples from Lake Magadi were analyzed for core lipid isoprenoid glycerol dialkyl glycerol tetraethers (iso-GDGTs) at the University of Massachusetts Amherst on an Agilent 1260 series high performance liquid chromatograph (HPLC) in tandem with an Agilent 6120 series single quadrupole mass selective detector (MSD). Compounds were ionized using atmospheric pressure chemical ionization

(APCI). The columns used for GDGT separation were a pre-column guard followed by two ultra-high performance liquid chromatography (UHPLC) silica columns (BEH HILIC, 2.1x150 mm, 1.7 μm,



Waters) connected in series and kept at 30 ºC. Elution solvents followed Hopmans et al. (2016) using a flow rate of 0.2 mL_min$^{-1}$. Two solvent mixtures, hexane (A) and 9:1 Hex:IPA (B), and were eluted isocratically for 25 minutes with 18% B, a linear gradient to 35% B in 25 minutes, a second linear gradient to 100% B in 30 minutes.

### 2.3.2   GDGT indices

Several different ratios based on the relative abundance of different isoprenoid GDGTs have been developed to determine their source(s). The methane index (MI) is defined by Zhang et al. (2011) and is calculated as in Eq. (1):

$$MI = \frac{GDGT-1+GDGT-2+GDGT-3}{GDGT-1+GDGT-2+GDGT-3+Cren+Cren'} \tag{1}$$

MI values range between 0 and 1 with values > 0.5 considered to be derived from methanotrophic communities and values < 0.3 considered normal sedimentary conditions (Zhang et al., 2011). These proposed ranges from Zhang et al. (2011) were derived from GDGTs found in marine sediments, so the cutoff values for methanotrophy may differ from lacustrine sediments, even those that are saline and alkaline.

The ratio of GDGT-2 / crenarchaeol (% GDGT-2 / cren) also indicates methanotrophy (values > 0.2), specifically methanotrophy associated with sulfate-methane transition zones (Weijers et al., 2011). These values were normalized and converted to percentages so that the numbers produced could be contextualized with the other indices used (Eq. 2). As a result, % GDGT-2 / cren contributions greater than 33% will be considered methanotrophic signals.

$$\%GDGT-2/\text{cren} = [GDGT-2]/([GDGT-2]+[Cren]) \; x \; 100 \; \% \tag{2}$$

Methanogenic inputs are calculated similarly to Eq. 2 above using GDGT-0 in place of GDGT-2. Blaga et al. (2009) found that values of GDGT-0 / (GDGT-0 + cren) > 2 are associated with methanogenic



archaeal communities in a study of freshwater lakes. Similarly, in a study of Eocene aged marine
sediments, Inglis et al. (2015) normalized the equation and converted it to a percentage, a convention we
follow (Eq. 3). They found that contributions from methanogens were indicated by values greater than
67%.

$$\% \; GDGT-0/\text{cren} = [GDGT-0]/([GDGT-0]+[Cren]) \; x \; 100 \; \% \tag{3}$$

The GDGT-2 / GDGT-3 ([2] / [3]) index was used here to describe both mesophilic environments as well
as environments with high MI values. This is the same version described in Rattanasriampaipong et al.
(2022) (Eq. 4).

$$[2]/[3] = [GDGT-2]/[GDGT-3] \tag{4}$$

$$\tag{5}$$

Carbon preference indices (CPI) were calculated for both the free fatty acid methyl esters (CPI_{FA}; Eq. 6)
and $n$-alkanes (CPI_{alk}; Eq. 7) to determine the source of leaf waxes and possible degradation of each wax
type. The equation for the CPI_{alk} is based on the work of Marzi et al. (1993) with an adaptation for the
CPI_{FA} from Naraoka and Ishiwatari (1999). Average chain length (ACL_{(25-33)}; Eq. 8) for $n$-alkane leaf
waxes with chain lengths of 25-33 carbons was also calculated based on the equation from Freeman and
Pancost (2014).

$$CPI_{FA} = 2 * \frac{\sum even \; C_{20} \; to \; C_{28}}{(\sum odd \; C_{19} \; to \; C_{27}) + (\sum odd \; C_{21} \; to \; C_{29})} \tag{6}$$

$$CPI_{alk} = \frac{1}{2} * \frac{(\sum odd \; C_{23} \; to \; C_{31}) + (\sum odd \; C_{25} \; to \; C_{33})}{\sum even \; C_{24} \; to \; C_{32}} \tag{7}$$

$$ACL_{(25-33)} = \frac{1}{2} * \frac{(25*[C_{25}] + 27*[C_{27}] + 29*[C_{29}] + 31*C_{31} + 33*[C_{33}]}{([C_{25}] + [C_{27}] + [C_{29}] + C_{31} + [C_{33}])} \tag{8}$$





# 3 Results

## 3.1 GDGT lipid variability

Samples are split into six intervals (1-6) based partly on their fractional abundances of GDGT-0 and cren as well as their MI values: (1) 35.668 to 32.61 m; ca. 17.7 to 14.9 ka, (2) 67.815 to 43.51 m; ca. 129 to 38.9 ka (3) 86.065 to 70.78 m; 197 to 149 ka, (4) 96.38 to 94.915 m; ca. 318 to 315 ka, (5) 104.10 to 103.16 m; ca. 324 to 323 ka, and (6) 130.21 to 119.64 m; ca. 456 to 391 ka (Table 1).

Fractional abundances of GDGTs along with sample depth and age are shown in Table 1 for each interval of the MAG-14-2A core demarcated by the respective sampling interval. GDGT-0 to -3 are present in addition to crenarchaeol and its regioisomer (hereafter, cren and cren'). Downcore GDGT ratios (MI, %GDGT-2 / cren, % GDGT-0 / cren and [2] / [3]) are summarized along with values of $\delta^{13}C_{OM}$ in Fig. 3 and are plotted against age (Fig. 3 and Table 1 for age/depth relationships). MI, % GDGT-0 / cren, and % GDGT-2 / cren values oscillate between high and low values, changing abruptly throughout the core. The methanotrophic (% GDGT-2 / cren) and the methanogenic (% GDGT-0 / cren) indices track similarly to MI values; that is, when values of MI are high, so are the other two indices. It should be noted that there are some large gaps in sampling between intervals in the core as a result of our sampling regime (i.e. targeting intervals with high apparent organic matter based on darker silty matrix).

Interval 1 is characterized by a higher proportion of cren and lower overall index values. The % GDGT-0 / cren index averages 36.3 % in this interval while the % GDGT-2 / cren index averages 10.6 % (Fig. 3). The MI in this interval is correspondingly low with an average of 0.25, well below the MI = 0.5 cutoff range for methanotroph-impacted communities. As such, this interval could be used for the [2] / [3] index; values averaged 2.1. Interval 2 has much higher values for each of these indices, where the average % GDGT-0 / cren = 99.3 % and the average % GDGT-2 / cren = 93.6 %. MI values in Interval 2 are also high with an average of 0.96. Of note, there is a large gap where no measurements were taken from 43.55 to 46.6815 m (~9.7 kyr) as well as from 50.36 to 58.74 m (~32.5 kyr). Interval 3 averages for % GDGT-0 / cren and % GDGT-2 / cren are 54.3 and 20.4 %, respectively. However, there is one anomalously high value at 77.32 m with % GDGT-0 / cren and % GDGT-2 / cren values at 99.6 and 93.8 % and an MI =



0.96. Excluding the high index value, the averages were lowered to 48.6 and 11.2 % for the % GDGT-0 / cren and % GDGT-2 / cren values and the MI average was lowered from 0.33 to 0.26. With the exclusion of 77.32 m, the [2] / [3] index averaged 1.5 in this interval, lower than Interval 1. Interval 4 is characterized by high index values, with a similarly abrupt shift from low values. Averages of the % GDGT-0 / cren and % GDGT-2 / cren are 98.1 and 88.2 % and an average MI of 0.92; these average index values are similarly high as compared to Intervals 2 and 6. Interval 5 is a shift to lower overall index values with averages of  % GDGT-0 / cren and % GDGT-2 / cren at 40.1 and 9.2 % and an average MI of 0.22. Finally, Interval 6 shows a period in the core with high index values throughout. Averages of % GDGT-0 / cren, % GDGT-2 / cren, and MI are 97.6 %, 89.4 %, and 0.95, respectively.

### 3.2  Bulk $\delta^{13}C_{OM}$ values

Values of bulk $\delta^{13}C_{OM}$ values are shown in Fig. 3 and Table 1. Bulk $\delta^{13}C_{OM}$ values follow a similar pattern to the indices described in section 3.1. Samples in Interval 1 ranged from -21.9 to -16.8 ‰ and had an average $\delta^{13}C_{OM}$ value of -18.4 ‰ with respect to VPDB. Interval 2 samples had the most $^{13}C$-depleted values in all sampled intervals, ranging from -89.4 to -24.7 ‰ with an average of -35.1 ‰. In Interval 3, the $\delta^{13}C_{OM}$ had a narrower range from -24.4 to -21.4 ‰ and an average of -22.5 ‰. A lighter signal from Interval 4 yielded a narrow range of values from -27.0 to -25.4 ‰ averaging -26.0 ‰. Interval 5 had slightly heavier values ranging from -25.0 to -18.1 ‰ with an average of -22.1 ‰. Lastly, Interval 6 had depleted $\delta^{13}C_{OM}$ values similar to intervals 2 and 4, with a range of -28.2 to -22.1 ‰ and an average of -25.2 ‰.

### 3.3  Leaf wax distributions

Only 15 samples were analyzed for FAMEs as they were part of pilot sampling and were not further expanded due to time constraints. Analyzed FAMEs exhibit an even over odd predominance, which is diagnostic of a primarily terrestrial source. $C_{16:0}$ to $C_{34:0}$ FAMES were present in the 15 samples analyzed with $C_{34:0}$ and $C_{24:0}$ averaging the highest percent abundance at 16.5 and 11.5 %, respectively, followed by $C_{17:0}$ and $C_{16:0}$ with averages of 10.8 and 10.3 %, respectively (Table S1). The $CPI_{FA}$ ranged from a





minimum of 1.1 to a max of 8.5 with an average of 3.82 in the 15 samples, indicating an overall terrestrial
source of FAMEs.

The *n*-alkanes represent a more robust sampling interval (n=56) that closely matches the total number of
GDGT sampling intervals (n=58). The *n*-alkanes analyzed showed an odd over even predominance, with
higher overall $ACL_{(25-33)}$ values indicating primarily terrestrial sources and no evidence for *n*-alkanes $C_{17}$
to $C_{22}$. Overall, $C_{31}$ and $C_{33}$ *n*-alkanes accounted for 54.4 % of all *n*-alkanes in every interval and were
particularly dominant in Intervals 2 through 6, while *n*-alkanes in Interval 1 were more evenly distributed,
though $C_{31}$ is still more dominant averaging 14.5%. The average value for the $CPI_{alk}$ in Interval 1 was 2.0
and the $ACL_{(25-33)}$ was 29.2. Intervals 2, 3, and 4 were much higher than Interval 1 with average values
of the $CPI_{alk}$ at 6.7, 7.0, and 6.1, respectively, while the values for the $ACL_{(25-33)}$ were 30.7, 30.9, and
30.3, respectively. Intervals 5 and 6 had similar values with the $CPI_{alk}$ averaging 4.6 and 4.2, respectively,
and the $ACL_{(25-33)}$ averaging 29.7 and 30.1 respectively.

In order to understand proportions of sulfate reducing bacteria (SRB), we focus here on the short-chain
length FAMEs. The short-chain $C_{15:0}$ FAMEs were not observed in Magadi sediments, possibly due to
degradation of the FAMEs smaller than $C_{16:0}$. These short-chain FAMEs are diagnostic of bacteria with
different respiratory pathways and are useful in understanding bacterial contributions in sedimentary
environments (Cho and Salton, 1966; Parkes and Taylor, 1983). Among bacterially diagnostic short-chain
FAMEs, $C_{15:0}$, $C_{15:0-iso}$, $C_{17:0}$, and $C_{17:0-iso}$ are used to identify SRB in sediments (Boon et al., 1977; Ueki
and Suto, 1979; Parkes and Taylor, 1983). In Magadi, only 15 samples were analyzed for FAMEs, and of
those 15 samples, only 22.8% of samples in Magadi contained $C_{17:0}$ fatty acids indicative of SRB. In
Interval 1, only one sample contained $C_{17:0}$ at ca. 14.9 ka with a concentration of 6.2 ng g sed$^{-1}$. Interval
2 had higher concentrations of $C_{17:0}$ with a range from 8.8 to 469.0 ng g sed$^{-1}$ and an average of 97.1 ng g
sed$^{-1}$. Similar to interval 1, interval 3 had only one sample with $C_{17:0}$ present at 390.8 ng g sed$^{-1}$, coinciding
with the high MI value in that interval at ca. 185 ka. Interval 4 $C_{17:0}$ values were similarly sparse, only
having two values at 8.3 and 73.9 ng g$^{-1}$ sediment extracted. In Interval 6, three values of $C_{17:0}$ were
recorded with values of 5.9, 31.5, and 359.6 ng g$^{-1}$ sediment extracted.



## 3.4 Bulk geochemistry

Bulk geochemical data and core descriptions from both Owen et al. (2019) and Owen et al. (2024) were also used to interpret hot spring influences in the intervals of focus (i.e., Intervals 1, 3, and 5). Both a PCA and correlation matrix of the MI and [2] / [3] compared to the rare earth elements (REEs) La, Ce, Nd, Sm, Eu, Tb, Yb, and Lu (Fig. 5). Increased values of REEs are characteristic of sodic systems influenced by hydrothermal springs, namely Mono Lake in California and this system (Johannesson and Lyons, 1994; Owen et al., 2019). A PCA (Fig. 5a) and non-parametric Spearman correlation matrix (Fig. 5b) were performed to quantify the relationship between REEs, MI, and [2] / [3]. The PCA showed that MI and [2] / [3] loaded positively on PC1, while the REEs Lu, Yb, Tb, Eu, and SM loaded positively on PC2 and the REEs Ce, Nd, and La loaded negatively on PCs 1 and 2. This indicates a negative relationship between a high MI and [2] / [3]. Similarly, the correlation matrix of REEs, MI, and [2] / [3] showed a negative relationship between each index and REE, except for the relationship of [2] / [3] and Nd, which showed no trend (r = 0.02).

## 4 Discussion

### 4.1 Lake Magadi microbial community shifts

The abrupt changes in isoprenoid GDGT-based indices in the sediment record of Lake Magadi indicate shifts in the archaeal communities present (Fig. 3) Shifts between two distinct communities were inferred using a combination of the Methane Index (MI) and ratios of GDGT-0 and GDGT-2 normalized to crenarchaeol (Eqs. 2 and 3; GDGT structures in Fig. 2). We denote these shifts as either "MI-on periods", characterized by MI > 0.5 during intervals 2, 4, and 6, and "MI-off periods", characterized by 0.3 < MI < 0.5 during intervals 1, 3, and 5. Oscillations between these two environmental states are discussed in detail in the following sections.

### 4.1.1 MI-on periods

In Lake Magadi, during the MI-on periods (Fig. 3; Intervals 2, 4, and 6), the MI is persistently greater than 0.83 and displays more $^{13}$C-depleted $\delta^{13}C_{OM}$ values compared to MI-off periods, indicating periods



of enhanced methane cycling. AOM is a likely mode of methane cycling in Interval 6 as well as parts of Interval 2 because SRB and AOM archaea live in a consortium together at the sulfate methane transition zone, or SMTZ (Boetius et al., 2000; Hinrichs and Boetius, 2002; Werne et al., 2004), and biomarkers of 350 SRB (FAMEs) were identified in those intervals. Thus, in intervals of the Magadi core where a SMTZ is suspected, such as in parts of Interval 2 and most of Intervals 4 and 6, there should be an increase in indices related to methanotrophy such as high MI and % GDGT-2 / cren (Weijers et al., 2011). Additionally, whereas methanogens and methanotrophs appear to be present in a consortium based on both the methane indices as well as bulk $\delta^{13}C_{OM}$, the majority of the contributions are coming from 355 methanogens as seen in the ternary plot in Fig. 6. This may seem counter-intuitive as the MI has been typically used to describe samples exhibiting a high predilection towards methanotrophy, but a high MI value does not necessarily exclude methanogenesis and conversely neither does a low MI, rather the low MI value suggests a predominance of Thermoproteota over Euryarchaeota (Zhang et al., 2011). High % GDGT-0 / cren and % GDGT-2 / cren index values in Intervals 2, 4, and 6 (Fig. 3) show that 360 methanogenesis is co-occurring with AOM. The [2] / [3] index is also useful in understanding the proportion of methanotrophs in sediments, even in intervals with high MI values like those discussed herein (Table 1; Fig. 3). Values of the GDGT [2] / [3] ratio track nearly identically to the MI values (Fig. 3), indicating that the MI is influenced by GDGT-2, which is characteristic of methanotrophs (Pancost et al., 2001; Schouten et al., 2003; Zhang et al., 2011).

Typically, methanogenesis in sulfate-rich systems is suppressed in favor of sulfate reduction caused by competition for both $H_2$ and organic substrates (Fazi et al., 2021; Sorokin et al., 2015). However, reports of methanogenesis co-occurring with SRB have been noted when methanogens are using non-competitive substrates such as methanol, or when sulfate levels are low (Oremland et al., 1982; Giani et al., 1984 370 Hoehler et al., 2001; Bebout et al., 2004; Arp et al., 2008, 2012; Jahnke et al., 2008; Smith et al., 2008; Robertson et al., 2009). Furthermore, pyrite nodules occur in most of the intervals where high index values are observed, indicating that there was a substrate for SRB, though it may have been in low concentration (Table 1). SRB can also be traced with FAMEs, specifically odd, short-chained FAMEs such as $C_{15:0}$ and $C_{17:0}$ in addition to their iso- and ante-iso forms (Boon et al., 1977; Ueki and Suto, 1979; Parkes and



Taylor, 1983). High relative abundance of $C_{17:0}$ (Tables 1 and 2) in the same intervals characterized by high index values suggest that methanotrophy is occurring in these sediment intervals. Thus, the combined evidence of higher FAMEs and pyrite nodules in intervals with high GDGT-based indices (e.g. MI, % GDGT-0 / cren, and % GDGT-2 / cren [2] / [3]) indicates the presence of a SMTZ that supports AOM with the co-occurrence of methanogenesis.

Interval 2 (Figs. 3 and 5) of the Magadi core appears to be more influenced by methanogenesis than AOM, resulting from a more prevalent % GDGT-0 / cren signal accompanied by a high % GDGT-2 / cren signal, high [2] / [3] ratios, and a more $^{13}$C-depleted bulk $\delta^{13}C_{OM}$ signal (average = -35.1 ‰; median = -28.6 ‰). Values of bulk $\delta^{13}C_{OM}$ are similarly $^{13}$C-depleted in AOM-dominant Euryarchaeotal systems ranging from

active mud volcanoes (~-27 ‰; ANME-1), a Danish freshwater lake (average ~-29.7‰; ANME-2), and the Sea of Galilee (~-30 ‰; ANME-2) in Israel (Lee et al., 2018; Norði et al., 2013; Sivan et al., 2011). However, when looking at Fig. 6, it appears that GDGT-0 is the dominant GDGT compared to GDGT-2 and cren, indicating that this interval is likely methanogen-dominant rather than ANME dominant. At points where the bulk $\delta^{13}C_{OM}$ values are at their lowest (e.g., -89 ‰), they are accompanied by a lower %

GDGT-2 / cren at ca. 95 % and an elevated GDGT-0 / cren at > 99.5 %. This is in line with the literature as Summons et al. (1998) reported values between -53.4 and -48.7 ‰ in the total lipid extract of methylotrophic methanogens using non-competitive substrates in anoxic hypersaline environments. Furthermore, as these waters are typically sulfate limited, it is likely that acetoclastic and/or hydrogenotrophic methanogenesis is dominant when evidence for SRB is lacking (i.e., pyrite, $C_{17:0}$

FAMEs). Zhuang et al. (2016) performed compound specific isotope analysis on several archaeol compounds from the Orca Basin and found archaeol and hydroxyarchaeol using $H_2$ or $CO_2$ (diagnostic of methanogens and methanotrophs) were relatively depleted (ca. -80 to -60 ‰) compared to the bulk $^{13}C_{OM}$ (ca. -22 ‰). Zhuang et al. (2016) concluded that acetoclastic and/or hydrogenotrophic methanogenesis was unlikely due to high $SO_4^{2-}$ concentrations in the Orca Basin, which may be the case in Lake Magadi.

In Interval 2, there is no evidence of pyrite and a limited number of samples with $C_{17:0}$ FAMEs, which indicates there may be other Euryarchaeotal communities with different forms of AOM occurring in the sediments. These other forms of AOM include nitrate/nitrite reduction and iron coupled to AOM (in 't



Zandt et al., 2018). This is further bolstered by the evidence outlined by Kim and Zhang (2023) that not only quantitatively linked AOM to high MI values, but also to non-Group I ANME Euryarchaea because other non-GDGT producing ANME (e.g. ANME-2 and ANME-3) were shown to co-exist with Group I ANME. In the intervals that are missing pyrite (i.e., most of Interval 2; see Table 1; Ferland, 2017), the pyrite nodules may have either been too small to see with the naked eye, or the excess $H_2S$ could have been incorporated into the kerogen by reacting with labile organic matter. As for the dearth of FAMEs observed, save for the samples at 58.80 and 58.74 m, there is limited evidence for sulfate reduction. From 59.40 to 58.80 m, values of the bulk $\delta^{13}C_{OM}$ dip as low as -89.4 ‰ (Table 1; Fig. 3), which aligns well with methanogenic archaeal biomass (Norði et al., 2013). However, as discussed above there is likely acetoclastic and/or hydrogenotrophic methanogenesis co-occurring in these high index intervals and is likely the dominant process where sulfate-dependent AOM is absent, and the sulfate-dependent AOM is likely replaced by a coupling to either nitrate/nitrite or iron reduction.

Samples in Interval 4 (Table 1) of the Magadi core have high index values, but no evidence for sulfate-dependent methanotrophy except for high MI values. These intervals are thus interpreted as being methanogenic, rather than methanotrophic. It is interesting to note the abundance of pyrite in the four samples with low MI values (Table 1; 104.10 to 103.16 m), indicating sulfate reduction not linked to AOM. This is not observed in any other location of the core and a hypothesized series of reactions is described below, which may be linked to an abundance of SRB, anaerobic ammonium oxidizing (anammox) bacteria, and Thermoproteota (aerobic ammonia oxidizing archaea, AOA) in the overlying water column. Due to periodic influxes of freshwater in Magadi, in addition to a permanent meromixis present in virtually all samples post 380 ka, the water column would have been oxic in the upper portion and anoxic below the chemocline. Freshwater pulses would have also brought nutrients to the lake such as ammonia ($NH_4^+$) and sulfate ($SO_4^{2-}$). The oxic portion of the water column would have supported microaerophilic AOA that oxidize $NH_4^+$ to nitrite ($NO_2^-$), which is then transported to the anoxic part of the water column (Straka et al., 2019). Here, anammox bacteria are using excess $NH_4^+$ and the $NO_2^-$ from the AOA and converting these to $N_2$. Excess $SO_4^{2-}$ is simultaneously being used by SRB creating $HS^-$ that is reacting with iron species in the sediments and being buried as pyrite. Ladderane lipids characteristic



of marine annamox bacteria (Jetten et al., 2009) were not studied in Magadi sediments. However, there is both 16S rRNA and lipid evidence for production of ladderanes in hot springs in the western United States suggesting that annamox bacteria can persist in hot spring environments (Jaeschke et al., 2009). Additionally, Kambura et al. (2016) found evidence for *Planctomycetes* in both mat and water samples
surrounding the hot springs of Lake Magadi lending credence to the hypothesis of AOA persisting in Lake Magadi. Without other lines of evidence, however, these are simply hypothesized reactions for explaining excess pyrite in the sediments without accompanying MI values, yet the explanation has some merit because of the high relative abundance of both crenarchaeol and cren'.

In nearly all of Interval 6 (Table 1), there is evidence for a higher proportion of methanotrophic archaea from 128.74 to 119.64 m (increased % GDGT-2 / cren and [2] / [3]) and methanogenesis in the intervals from 130.21 to 129.77 m (Table 1; higher % GDGT-0 / cren compared to % GDGT-2 / cren). Samples from 123.43 to 119.64 (Table 1) are of note because the [2] / [3] values are lower than the MI values whereas every other MI and [2] / [3] values aligned nearly 1:1. This is likely due to GDGT-2 not being
the dominant control of the MI and while both % GDGT-0 / cren and % GDGT-2 / cren are equally high, there may be other factors in the water column exporting GDGT-2 to the sediments, possibly from deep-dwelling Group I.1b Thermoproteota, although this is unlikely due to limitations of depth (Taylor et al., 2013). The 656 m paleoshoreline reported by Casanova (1986, 1987) and Casanova and Hillaire-Marcel (1987) would imply a maximum water depth of ~ 50 m during the Late Pleistocene (African Humid
Period: AHP) based on present topography. However, earlier water depths are unclear because accommodation space was changing as the axial rift developed with faulting and subsidence (Owen et al., 2024). This is not deep enough (> 1 km) to support the Group I.1b Thermoproteota per the constraints outlined in Taylor et al. (2013).

### 4.1.2 MI-off periods

In intervals characterized by low MI, % GDGT-0 / cren and % GDGT-2 / cren values (Fig. 3; MI-off intervals outlined in green checkered patterns), the $\delta^{13}C_{OM}$ values are $^{13}C$-enriched relative to those intervals characterized by higher index values (MI-on) (Fig. 3). Since the methane cycling indices (%



GDGT-0 / cren and % GDGT-2 / cren) are both predominantly influenced by the availability of crenarchaeol, MI-off periods are marked by an increased production in crenarchaeol. Typically,

crenarchaeol is produced in open ocean systems, freshwater lakes, and soils by the mesotrophic aerobic ammonium oxidizing phylum Nitrososphaerota. However, they can also be found in other environments such as hot spring mats in Thermoproteota (Pearson et al., 2004, 2008; Schouten et al., 2013). As Thermoproteota require oxygen to oxidize ammonium to nitrate, the increased presence of crenarchaeol in the MI-off intervals therefore suggests periods of time during which conditions were more oxic, at least

in the upper water column.

The MI-off periods in Lake Magadi are the core intervals where there is an increase in the relative abundance of crenarchaeol in the sediments, driving the MI below the 0.5 threshold that defines methanotrophy (Zhang et al., 2011). The low MI values are accompanied by equally low values in the

other indices and relatively [13]C-enriched $\delta^{13}C_{OM}$ values (Fig. 3). The increase in crenarchaeol as well as the low [2] / [3] index values suggest that more Thermoproteota are present in Magadi in these periods. As mentioned in Section 4.1.1, three groups of AOA are of interest for interpreting what archaeal groups are found in low index intervals of Lake Magadi. Averages of [2] / [3] from the global dataset in Rattanasriampaipong et al. (2022) are as follows: hot spring mats (avg. = 1.00), shallow AOA cultures

(avg. = 1.16), and shallow core tops (avg. = 2.64). Placing these on a continuum, we can approximate the environment from [2] / [3] averages in Magadi, though it should be noted that the shallow AOA and shallow core top values in Rattanasriampaipong et al. (2022) are based on marine core tops, while the hot spring mats are based on terrestrial hot springs like those observed around Lake Magadi (i.e., pH > 6.5).

Interval 1 captures a transition from a more arid East Africa to a wetter period at the onset of the African Humid Period (AHP). During wetter periods, more allochthonous material is carried into the lake, which includes vegetation that impacts the overall bulk $\delta^{13}C_{OM}$ values. This allochthonous vegetation enriches the overall bulk $\delta^{13}C_{OM}$ values more significantly compared to other intervals in the Magadi core. Average values of bulk $\delta^{13}C_{OM}$ are -17.7 ‰ in Interval 1, which correspond to the $\delta^{13}C_{OM}$ values of aquatic sedges

mixed with a terrestrial signal of grassy woodland (Sikes, 1994; Reiffarth et al., 2016). The average





ACL$_{(25-33)}$, CPI$_{alk}$, and CPI$_{FA}$ values in Interval 1 are 29.4, 2.0, 3.2 respectively, suggesting a C4 vegetation origin of the long-chain $n$-alkanes as indicated by the value of 29.4 and moderate degradation overall as implied by the low CPI$_{alk}$ and CPI$_{FA}$ values (Table S1). This degradation is likely why no $n$-alkanes shorter than C$_{22}$ were identified in the core. Furthermore, pollen records in Lake Magadi indicate that a mixture

of C4 grassy woodlands and C4 aquatic sedges were predominant in the landscape that surrounded Lake Magadi at this time (Muiruri et al., 2021). Supporting the pollen record, the δ$^{13}$C$_{OM}$ values are likely reflecting δ$^{13}$C values similar to those observed by Garcin et al. (2014) in equatorial regions of Cameroon. The bulk δ$^{13}$C$_{OM}$ is likely recording a mixture of C4 grasses and C4 sedges similar to δ$^{13}$C values of C$_{27}$ to C$_{33}$ $n$-alkanes obtained from C4 grasses and sedges in Cameroon which ranged from -18.2 to -17.6 ‰

(Garcin et al., 2014). This all suggests that the bulk δ$^{13}$C$_{OM}$ signal is dominated by terrestrial biomass, and there does not appear to be a significant influence from the benthic microbial community (i.e., methane cyclers or SRB).

Values of the [2] / [3] index average 2.1 in Interval 1 with some values as high as 3.74 and 4.63 at 33.28

and 33.03 m, respectively (Table 1). The higher values are closer to what is captured from deep oceanic suspended particulate matter (SPM) and deep ocean core tops below the pycnocline, though caution should be used when comparing lacustrine and oceanic sediments (Rattanasriampaipong et al., 2022). The increase in % GDGT-0 / cren (50.6 and 54.3 %; Table 1) and the slightly increased MI values (0.37 and 0.41; Table 1) suggest these samples were deposited in a deeper lacustrine environment. Evidence for a

deeper paleolake at ~40 m above the modern lakeshore (Baker, 1958) is also observed in the High Magadi Beds (ca. 17.7 to 10.8 ka) indicating that there was freshwater flowing into the lake during the period of deposition in Interval 1, likely creating a freshwater cap on the meromictic Lake Magadi (Barker et al., 1991; Behr, 2002; Owen et al., 2019). However, excluding the high [2] / [3] index values in Interval 1, the average is 1.6, which is closer to the hot spring mats and shallow AOA cultures (Rattanasriampaipong

et al., 2022). Likely, the higher [2] / [3] index values mean periods of increased methanogenesis occurring in the sediments, with AOA input from the upper water column likely induced by increased hydrothermal activity (see Section 4.2.1). In the periods of lower [2] / [3] values, the community is interpreted as being dominated by AOA and thermophilic AOA cultures (i.e., Thermoproteota; Rattanasriampaipong et al.,





2022) and is further supported by high % cren and % cren'. Kumar et al. (2019) described similarly low

[2] / [3] values in the water column of Lake Malawi that are akin to values observed in Lake Magadi in

both Intervals 1 and 3. They found that values of a lower normalized [2] / [2+3], ranging from 0.55 to

0.59, in Lake Malawi were associated with the shallower Thermoproteota (Thaumarchaeota) Group I.1b.

This is compared to greater values of [2] / [2+3] in the deeper dwelling Thermoproteota Group I.1a, which

means that most samples in Interval 1 are likely sourced from Group I.1b (Kumar et al., 2019). The

interpretations of Kumar et al. (2019) concluded that Group I.1b Thermoproteota were contributing to the

lower [2] / [2+3] values, while the more deeply dwelling Group I.1a Thermoproteota were more prevalent

in aphotic portions of the water column (Kumar et al., 2019). The normalized [2] / [2+3] used by Kumar

et al. (2019) with values ranging from 0.55 to 0.65 approximates values of [2] / [3] in the 1.30 to 1.65

range as described in this paper. More recently, Baxter et al. (2021) found that Thermoproteota I.1b are

more prevalent in the upper oxygenated portion of the water column within the photic zone as evidenced

by a higher relative abundance of crenarchaeol and lower relative abundance of GDGT-2. Thus, our

interpretations of thaumarcheotal AOA in Lake Magadi sediments are consistent with data from Baxter

et al. (2021) and Kumar et al. (2019). This interpretation is consistent with Interval 1 being a period of

proportionately more freshwater and $HCO_3^-$-rich hydrothermal input and a deeper lake overall, which

would explain the accompanying increase in crenarchaeol.

Interval 3 [2] / [3] averages are lower overall (Table 1; avg. = 5.4), with only one outlying high value (ca.

77.32 m at a value of 36.7). Excluding this high index value, the [2] / [3] average drops to 1.5, which is

closer to what is observed in shallow AOA cultures and hot spring mats. With most samples being closer

to unity (i.e., [2] / [3] = 1.0), it is likely that hot springs had a greater influence on the community

composition in these intervals. Samples that are closer to unity (70.78, 70.86, and 71.08 to 75.93 m) also

have a relatively $^{13}$C-enriched $\delta^{13}C_{OM}$ values (avg. = -21.8‰) compared to samples with a higher [2] / [3]

(averaging 1.5 excluding the outlying value of 36.7). This average is closer to shallow Group I.1a

Thermoproteota as described previously. Average isotope values in Interval 3 are between oceanic

hydrothermal vents (avg. = -19.0 ‰) and terrestrial alkaline hot spring systems such as the Bison Pool

hot spring in Yellowstone National Park (avg. = -24.9 ‰) (Shah et al., 2008; Schubotz et al., 2013). Since



elevated amounts of GDGT-2 (i.e., relative abundance > 45%) are associated with Euryarchaeota, and values in Intervals 1, 3, and 5 are much lower than 45% (Table 1), these intervals are likely dominated by Thermoproteotal AOA (Pancost et al., 2001; Turich et al., 2007; Taylor et al., 2013). Archaeal community composition in Intervals 1, 3, and 5 is independent of these external factors and is related to hydrothermal flows. $CPI_{alk}$, $CPI_{FA}$, and $ACL_{(25-33)}$ average 7.0, 5.0, and 30.9, which indicates a higher terrestrial input that tracks with the aridity during this period. This further supports the hot springs driving the lake archaeal community composition as there was less overall precipitation and the Thermoproteotal communities were more abundant during Intervals 1, 3, and 5.

Lastly, in Interval 5, which only has 4 samples, has similarly low values of [2] / [3] (average = 1.4) like Intervals 1 and 3, which is likely indicative of Thermoproteotal AOA cultures. The $CPI_{alk}$ and $CPI_{FA}$ averages were 4.6 and 5.0 which indicates more of a terrestrial input. So, while these values are lower than Interval 3, and closer to the values in Interval 1, these still indicate a higher terrestrial input during this time frame.

## 4.2    GDGTs: A molecular "switch" recording environmental change due to hydrothermal influences

In the Lake Magadi core, there are intervals in which sediments have a tuffaceous quality  implying a more hydrothermal origin produced in situ (Fig. 3, data from Owen et al., 2019). Hydrothermal fluids in the basin are rich in carbonate and bicarbonate as well as $Na^+$ ions as a direct result of the weathering and alteration of trachytic (silica-rich) basement rock (Jones et al., 1977; Allen et al., 1989). Zeolites form in these intervals due to the interaction of trachytic glass with saline and alkaline lake and hydrothermal waters and may be evidence for a larger influx of geothermal fluids in the lake (Owen et al., 2019).

There is abundant evidence for hydrothermal influences in the sediments in the Lake Magadi core (Owen et al., 2019). Owen et al. (2019) described the Magadi core from the trachytic basement layer at ca. 1,056 ka to modern alkaline pan samples. Samples ranging from ~545 to 380 ka (Figs. 3 and 4; Interval 6) marked a drier period where the lake area and volume had shrunk, and lake floor anoxia was prevalent





(Owen et al., 2019). Sediments in Interval 6 do not show any evidence of hot spring activity (i.e., there is

a higher Ca / Na; Fig. 4), and the main zeolite in this interval is analcime, indicating more evaporitic

saline as well as alkaline waters (Owen et al., 2019). In the core intervals from ~380 to 105 ka (Fig. 3;

Interval 5 through mid-Interval 2), the system was flooded periodically with freshwater but maintained

meromixis and a high proportion of zeolites forming in the lake waters with analcime as the dominant

zeolite indicating a higher salinity (Owen et al., 2019). Starting at ca. 380 ka to modern day, hydrothermal

fluids with abundant $HCO_3^-$ and $CO_3^{2-}$ ions and $Na^+ > Ca^{2+}$ have heavily influenced both deposition and

chemical alteration of sediments (Owen et al., 2019). Evaporitic enrichment of alkaline hydrothermal

fluids circulating through trachytic basement rocks and the increase in their contribution to Lake Magadi

are responsible for the alteration of the volcanic glasses. An abundance of analcime, combined with the

$Na^+ > Ca^{2+}$, indicates a proportionally higher influx of hydrothermal waters in Intervals 5 through 2 (Fig.

4; Owen et al., 2019). Similarly, from ~105 – 0 ka (Fig. 3; mid-Interval 2 through Interval 1) there is

evidence of a low Ca / Na, increased Br, and abundant zeolite formation, namely analcime, pointing to

increased hydrothermal flow in this interval as well (Fig. 4; Owen et al., 2019). These hydrothermal flows

are especially prevalent (low Ca/Na values) at intervals where tuffaceous silts are the dominant lithology

of the core.

The interval of tuffaceous silts occur at 96 – 102 m core depth, and is marked by increases in zeolitic

alteration from saline water inflow and a low Ca/Na ratio (Fig. 4, Intervals 1, 3, and 5). While the

tuffaceous silts at 30 – 36 and 74 – 76 m were changed to muds in Renaut and Owen (2023), REE data

still indicate a hydrothermal origin of MI-off intervals (Fig. 5). Tuffaceous silt intervals align well overall

with the low-MI values; sampled material with low-MI values is found wholly within the tuffaceous silts,

indicating that these values are still likely related to increased hydrothermal inputs. The presence of chert

in non-tuffaceous samples might also be a hydrothermal indicator, as the circulating hydrothermal fluids

beneath Lake Magadi are silica-rich (Eugster, 1969). Periods of increased hydrothermal activity were

likely occurring throughout Lake Magadi's history as there is evidence for chert beds and nodules in the

Quaternary sediments surrounding the lake – namely the Oloronga, Green, and High Magadi Beds (Behr,

2002; Renaut and Owen, 2023). The chert-bearing Oloronga Beds were deposited from approximately




to 300 ka, while the Green Beds were actively precipitating chert from ~ 220 to 70 ka and more recently, the High Magadi Beds were being deposited between ~25 and 9 ka (Fairhead et al., 1972; Goetz

and Hillaire-Marcel, 1992; Williamson et al., 1993; Behr and Röhricht, 2000; Owen et al., 2019; Reinhardt et al., 2019). Thus, samples in the low-MI intervals (ca. 32.61 to 35.67, 70.78 to 75.93, and 103.2 to 104.1 m) are likely reflecting these periods of increased hydrothermal activity in Magadi and the surrounding landscape.

## 5. Conclusions

Sediments in Lake Magadi track the environmentally driven changes in archaeal communities over the past ~ 456 ka. Using the MI to track the predominantly microbial inputs at Lake Magadi, we have observed sudden and distinct shifts between mixed archaeal communities of Euryarchaeotal methanogens and methanotrophs transitioning to mesophilic AOA Thermoproteota communities and back again. This shift is driven, in part, by moisture balances in the East African Rift, with wetter conditions leading to

lake stratification, and with more archaea derived from the upper water column rather than the sediments, as evidenced by low MI, low [2] / [3], and relatively $^{13}$C-enriched bulk $\delta^{13}C_{OM}$. There is also a clear relationship between low MI values and enhanced hydrothermal activity whereby increased hydrothermal activity (as recorded by tuffaceous silts and REEs in the sediments) maintains low overall index values, indicating that environmental conditions supported more mesophilic Thermoproteota. In contrast, periods

of lower hydrothermal activity are connected to high index values. Thus, the MI indicates a switch between high and low hot spring activity in Lake Magadi. The intervals of MI-off periods coincide with the tuffaceous units of inferred high hot spring activity in the geologic record, while MI-on periods occur during periods of lower hydrothermal activity. As this was one of the first studies to look at methane cycling in a soda lake over geologic time, we have gained valuable insights into how variable these

systems can be. Soda lakes are important ecosystems for methane cyclers and should be studied more closely so that we can better understand global methane contributions both in the past and better constrain sources in the future.



## 6.  Competing interests

The contact author has declared that none of the authors has any competing interests.

## 7.  Acknowledgements

We thank the Kenya National Council for Science and Technology (NCSTI) for granting research permits. Drilling and environmental permits were provided by the Kenya Ministry of Petroleum Mining and the National Environmental Management Authority of Kenya. We especially thank the National Environment Management Authority (NEMA). We give special thank the local Magadi Township Maasai community

for their approval of the project and Tata Chemicals Magadi for providing, who provided field support. DOSECC Exploration Services supervised drilling that was undertaken by Drilling and Prospecting International (DPI). We also thank the CSDF facilities (University of Minnesota) for allowing us to store and log our cores at their repository. Drilling at Magadi for the Hominin Sites and Paleolakes Drilling Project (HSPDP) was funded by ICDP and NSF grants (EAR-1123942, BCS-1241859, EAR-1338553).

This is Publication #XXX of the Hominin Sites and Paleolakes Drilling Project.





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




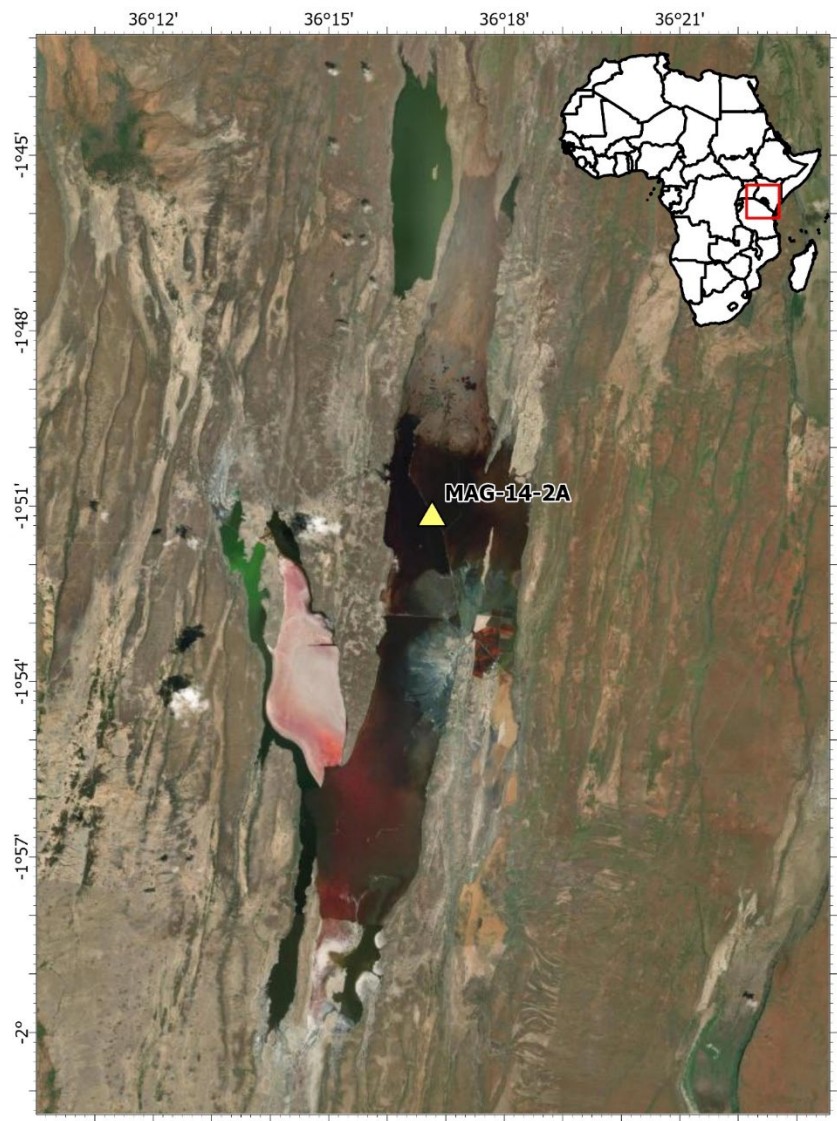


**Figure 1.** Map of the drilling location of MAG-14-2A (yellow triangle) in Lake Magadi for the Hominin Sites and Paleolakes Drilling Project (HSPDP). Credit to ESRI 2024.





**Figure 2.** Isoprenoid glycerol dialkyl glycerol tetraethers (GDGTs) structures, which are used in the various methane indices. Figure is based on Castañeda and Schouten (2011).





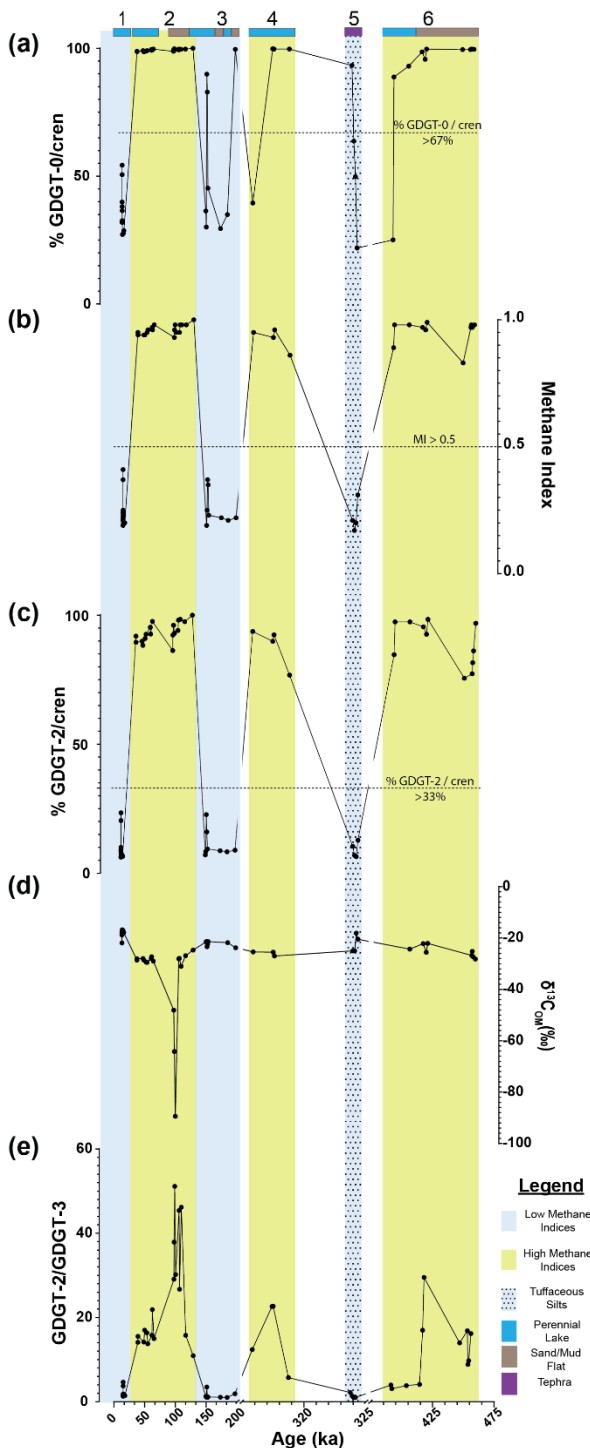


**Figure 3.** Downcore variations in Lake Magadi of the **a)** % 0 / Cren, **b)** MI, **c)** % 2 / Cren, **d)** bulk $\delta^{13}C_{OM}$, and **e)** the GDGT-2 / GDGT-3 ([2] / [3]) values from ca. 14.9 to 456 ka. Sections 1, 3, and 5 are low MI




intervals outlined in blue, the high MI intervals in Sections 2, 4, and 6 are in yellow. Checkered patterns
indicate periods of tuffaceous silt deposit, which align with the low MI intervals. Bands at the top of the
graph indicate the inferred (via Renaut and Owen, 2023) lake levels and major inputs with dark blue
indicating a perennial lake, brown indicating a sand or mud flat, and purple indicating tephra. Dotted lines
on each section denote the cut-off points for methane related indices MI (> 0.5), % GDGT-2 / cren (> 33
%), and % GDGT-0 / cren (> 67 %). See Section 2.3.2 for more details. Note the breaks in the X-axis
scale.

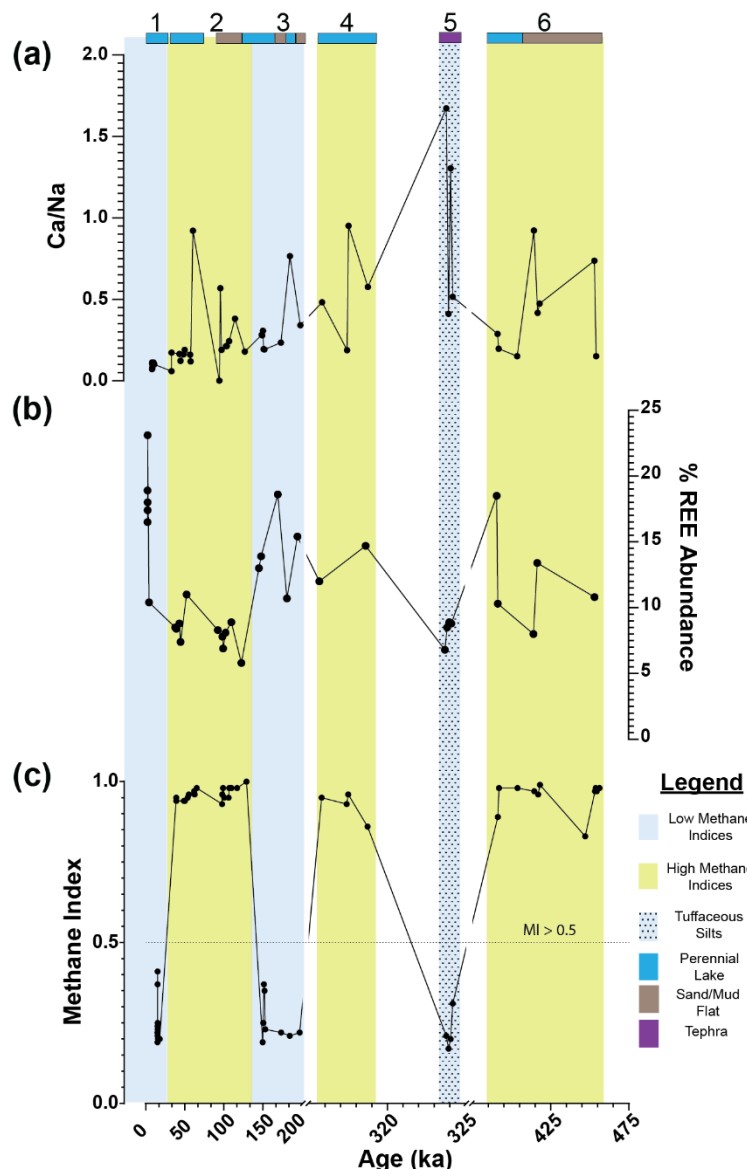






**Figure 4.** Downcore plot for Lake Magadi of **a)** Ca/Na, **b)** % REE abundance, and **c)** MI. Values range from ca. 14.9 to 456 ka and Sections 1, 3, and 5 are outlined in blue reflecting a low MI interval, while high MI intervals are outlined in yellow. The checkered pattern is indicative of periods of higher inferred hydrothermal flow. Bands at the top of the graph indicate the inferred (via Renaut and Owen, 2023) lake levels and major inputs with dark blue indicating a perennial lake, brown indicating a sand or mud flat, and purple indicating tephra. The dotted line on the MI plot **(c)** denotes the cutoff point > 0.5 for values significantly affected by methane cycling archaea. Note the breaks in the X-axis scale. REE values are from Owen et al. (2019).



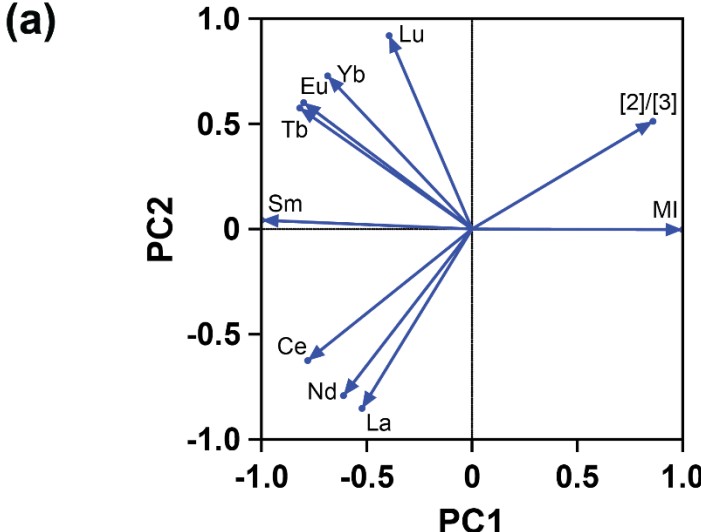

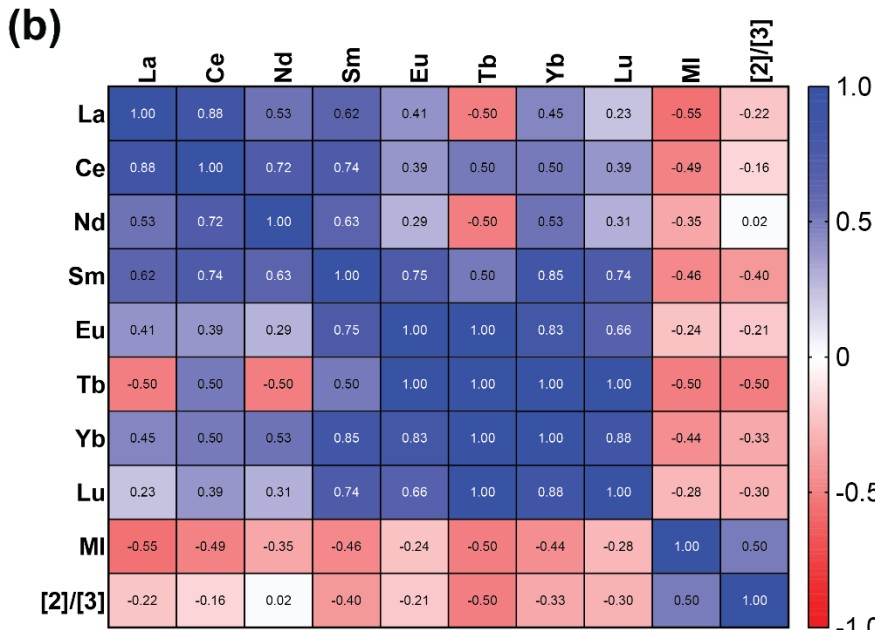

**Figure 5.** Both **a)** PCA and **b)** Spearman Correlation Matrix showing the relationship between methane related indices (MI and [2] / [3]) and REEs (La, Ce, Nd, Sm, Eu, Tb, Yb, and Lu) in the sampled intervals of the core. A negative relationship is seen between the methane indices and REEs as shown by opposing eigenvectors on the PCA **(a)** and negative r values on the correlation matrix **(b)**. REE values are from Owen et al. (2019).





**Figure 6.** Ternary plot of crenarchaeol, GDGT-0, and GDGT-2, which are used to calculate the methane indices. Samples are split by both their interval (denoted by their shape) and whether they are from a high MI (yellow) or low MI (blue) interval. Higher proportions of GDGT-0 indicate methanogenic inputs, higher GDGT-2 indicate methanotrophy, and higher crenarchaeol indicates more mesophilic conditions influenced by hot springs.



| | Depth (mbs) | Age (ka) | % GDGT-0 $m/z$=1302 | % GDGT-1 $m/z$=1300 | % GDGT-2 $m/z$=1298 | % GDGT-3 $m/z$=1296 | % Cren $m/z$=1292 | % Cren' $m/z$=1292 | % 0/Cren | % 2/Cren | MI | [2]/[3] | f[cren'] | Ca/Na | Bulk $\delta^{13}C_{OM}$ (‰) | Pyrite Present? | C17:0 ng g$^{-1}$ sed extracted |
|---|---|---|---|---|---|---|---|---|---|---|---|---|---|---|---|---|---|
| Interval 1 | 32.61 | 14.94 | 34.1% | 6.6% | 3.4% | 2.6% | 51.5% | 1.8% | 39.9% | 6.2% | 0.19 | 1.30 | 0.03 | 0.07 | -21.9 | - | 6.2 |
| | 32.79 | 14.96 | 26.3% | 7.8% | 5.4% | 3.2% | 55.8% | 1.5% | 32.0% | 8.8% | 0.22 | 1.66 | 0.03 | 0.11 | -18.5 | - | - |
| | 33.03 | 14.99 | 40.4% | 12.1% | 10.4% | 2.2% | 34.0% | 1.0% | 54.3% | 23.4% | 0.41 | 4.63 | 0.03 | 0.11 | -18.8 | - | - |
| | 33.28 | 15.02 | 38.5% | 10.8% | 9.6% | 2.6% | 37.6% | 0.9% | 50.6% | 20.4% | 0.37 | 3.74 | 0.02 | 0.07 | - | - | - |
| | 33.55 | 15.05 | 31.4% | 7.9% | 5.4% | 3.1% | 50.9% | 1.3% | 38.1% | 9.6% | 0.24 | 1.72 | 0.03 | 0.08 | -17.7 | - | - |
| | 33.76 | 15.07 | 27.0% | 7.5% | 5.0% | 3.2% | 55.7% | 1.6% | 32.6% | 8.3% | 0.21 | 1.57 | 0.03 | 0.08 | -18.3 | - | - |
| | 34.06 | 15.10 | 26.2% | 8.0% | 5.5% | 3.3% | 55.8% | 1.4% | 31.9% | 8.9% | 0.23 | 1.67 | 0.02 | 0.07 | -18.7 | - | - |
| | 34.15 | 15.10 | 29.5% | 8.7% | 5.7% | 3.2% | 51.4% | 1.5% | 36.5% | 10.0% | 0.25 | 1.82 | 0.03 | 0.08 | -18.1 | - | - |
| | 35.07 | 15.45 | 22.6% | 7.8% | 4.9% | 3.0% | 60.4% | 1.5% | 27.2% | 7.4% | 0.20 | 1.63 | 0.02 | 0.10 | -16.8 | - | - |
| | 35.42 | 16.81 | 23.0% | 7.8% | 4.6% | 2.8% | 60.3% | 1.6% | 27.6% | 7.1% | 0.20 | 1.67 | 0.03 | 0.11 | -17.2 | - | - |
| | 35.67 | 17.77 | 23.9% | 7.9% | 4.1% | 2.9% | 59.6% | 1.6% | 28.7% | 6.5% | 0.20 | 1.43 | 0.03 | 0.10 | -17.9 | - | - |
| Interval 2 | 43.51 | 38.99 | 78.5% | 8.9% | 10.9% | 0.8% | 1.0% | 0.1% | 98.8% | 91.9% | 0.95 | 14.13 | 0.07 | 0.06 | -28.6 | - | 11.5 |
| | 43.55 | 39.05 | 81.1% | 8.3% | 8.9% | 0.6% | 1.0% | 0.1% | 98.7% | 89.5% | 0.94 | 15.60 | 0.07 | 0.17 | -28.1 | - | - |
| | 46.68 | 48.70 | 85.2% | 6.7% | 6.8% | 0.5% | 0.8% | 0.1% | 99.1% | 89.9% | 0.94 | 14.21 | 0.09 | 0.17 | -28.0 | - | 9.3 |
| | 46.96 | 50.00 | 81.1% | 8.8% | 8.5% | 0.5% | 1.1% | 0.1% | 98.6% | 88.3% | 0.94 | 16.94 | 0.07 | 0.12 | -28.6 | - | - |
| | 47.75 | 53.61 | 81.0% | 8.2% | 9.3% | 0.6% | 0.9% | 0.1% | 98.9% | 91.0% | 0.95 | 16.26 | 0.08 | 0.16 | -29.4 | - | - |
| | 48.08 | 55.11 | 81.3% | 7.6% | 9.5% | 0.7% | 0.8% | 0.1% | 99.1% | 92.6% | 0.96 | 13.77 | 0.09 | 0.19 | -29.5 | - | - |
| | 49.55 | 61.83 | 83.7% | 6.8% | 8.4% | 0.5% | 0.4% | 0.1% | 99.5% | 95.3% | 0.97 | 15.92 | 0.19 | 0.16 | -28.0 | - | 26.8 |
| | 49.68 | 62.45 | 82.9% | 7.7% | 8.2% | 0.4% | 0.7% | 0.1% | 99.2% | 92.7% | 0.96 | 21.68 | 0.14 | 0.12 | -27.3 | - | - |
| | 50.36 | 65.22 | 79.0% | 9.0% | 10.9% | 0.7% | 0.3% | 0.1% | 99.7% | 97.6% | 0.98 | 14.88 | 0.24 | 0.92 | -29.0 | - | 57.2 |
| | 58.74 | 97.72 | 84.9% | 7.5% | 6.4% | 0.2% | 1.0% | 0.1% | 98.8% | 86.3% | 0.93 | 28.86 | 0.08 | 0.00 | - | - | 469.0 |
| | 58.80 | 97.97 | 88.3% | 6.0% | 5.1% | 0.1% | 0.4% | 0.0% | 99.5% | 92.3% | 0.96 | 39.23 | 0.07 | 0.00 | -48.1 | - | - |
| | 59.06 | 99.04 | 90.7% | 4.5% | 4.6% | 0.1% | 0.2% | 0.0% | 99.8% | 96.1% | 0.98 | 50.56 | 0.12 | 0.57 | -64.2 | - | - |
| | 59.40 | 100.46 | 92.6% | 2.3% | 4.5% | 0.2% | 0.3% | 0.1% | 99.6% | 93.0% | 0.95 | 29.87 | 0.16 | 0.19 | -89.4 | - | - |
| | 62.06 | 106.05 | 86.9% | 3.6% | 8.7% | 0.2% | 0.5% | 0.1% | 99.4% | 94.1% | 0.95 | 45.63 | 0.19 | 0.21 | -28.1 | - | - |
| | 62.65 | 106.93 | 82.0% | 3.7% | 13.4% | 0.5% | 0.3% | 0.1% | 99.7% | 98.1% | 0.98 | 26.88 | 0.35 | 0.21 | -27.9 | - | - |
| | 64.52 | 109.73 | 77.2% | 5.0% | 17.1% | 0.4% | 0.3% | 0.1% | 99.6% | 98.4% | 0.98 | 46.24 | 0.23 | 0.24 | -31.0 | - | - |
| | 65.98 | 116.93 | 84.1% | 3.6% | 11.2% | 0.7% | 0.3% | 0.0% | 99.7% | 97.5% | 0.98 | 15.77 | 0.11 | 0.38 | -26.9 | - | 8.8 |
| | 67.82 | 129.05 | 90.5% | 4.4% | 4.7% | 0.4% | 0.0% | 0.0% | 100.0% | 100.0% | 1.00 | 10.91 | | 0.18 | -24.7 | - | - |
| Interval 3 | 70.78 | 149.74 | 30.9% | 6.0% | 4.2% | 3.3% | 54.1% | 1.5% | 36.4% | 7.1% | 0.19 | 1.26 | 0.03 | 0.28 | -21.5 | - | - |
| | 70.86 | 150.43 | 23.6% | 9.3% | 5.1% | 4.6% | 54.9% | 2.6% | 30.1% | 8.4% | 0.25 | 1.09 | 0.05 | 0.28 | -21.4 | - | - |
| | 70.97 | 151.29 | 84.4% | 2.3% | 2.8% | 0.8% | 9.4% | 0.3% | 89.9% | 22.7% | 0.37 | 3.48 | 0.03 | 0.31 | -23.4 | - | - |
| | 71.08 | 152.17 | 75.4% | 2.8% | 3.0% | 3.0% | 15.5% | 0.4% | 82.9% | 16.0% | 0.35 | 1.00 | 0.03 | 0.19 | -22.4 | - | - |
| | 71.19 | 153.06 | 38.2% | 5.4% | 4.8% | 4.3% | 46.1% | 1.2% | 45.3% | 9.4% | 0.23 | 1.12 | 0.03 | 0.19 | -21.4 | - | - |
| | 73.70 | 173.42 | 24.2% | 5.9% | 5.5% | 5.2% | 57.9% | 1.4% | 29.5% | 8.7% | 0.22 | 1.07 | 0.02 | 0.23 | - | - | - |
| | 75.93 | 184.48 | 29.3% | 5.3% | 4.9% | 4.9% | 54.3% | 1.3% | 35.0% | 8.3% | 0.21 | 1.00 | 0.02 | 0.77 | -21.8 | - | - |
| | 77.32 | 185.23 | 90.1% | 3.9% | 5.5% | 0.2% | 0.4% | 0.1% | 99.6% | 93.8% | 0.96 | 36.73 | 0.12 | 0.34 | -24.4 | Y | 390.8 |
| | 86.07 | 197.23 | 33.4% | 6.7% | 5.0% | 2.6% | 51.0% | 1.3% | 39.5% | 8.9% | 0.22 | 1.90 | 0.03 | 0.48 | -23.8 | - | - |
| Interval 4 | 94.92 | 315.79 | 95.3% | 1.2% | 3.0% | 0.2% | 0.2% | 0.0% | 99.8% | 93.7% | 0.95 | 12.54 | 0.08 | 0.19 | -25.4 | - | 8.3 |
| | 95.05 | 317.40 | 95.7% | 1.3% | 2.6% | 0.1% | 0.3% | 0.0% | 99.7% | 89.9% | 0.93 | 23.45 | 0.09 | 0.95 | -25.5 | - | - |
| | 95.15 | 317.50 | 92.4% | 3.5% | 3.7% | 0.2% | 0.3% | 0.0% | 99.7% | 92.4% | 0.96 | 22.81 | 0.09 | 0.58 | -27.0 | - | - |
| | 96.38 | 318.74 | 65.5% | 11.5% | 15.6% | 2.7% | 4.7% | 0.0% | 93.3% | 76.8% | 0.86 | 5.73 | 0.00 | 1.67 | - | - | - |
| Interval 5 | 103.16 | 323.77 | 57.3% | 3.4% | 3.8% | 1.7% | 32.7% | 1.1% | 63.7% | 10.4% | 0.21 | 2.18 | 0.03 | 0.41 | -25.0 | Y | 73.9 |
| | 103.49 | 323.92 | 44.2% | 3.9% | 3.3% | 2.4% | 44.5% | 1.7% | 49.8% | 6.9% | 0.17 | 1.40 | 0.04 | 1.30 | -25.0 | Y | - |
| | 103.79 | 324.05 | 17.6% | 7.7% | 4.3% | 4.4% | 62.8% | 3.2% | 21.9% | 6.4% | 0.20 | 0.97 | 0.05 | 0.52 | -18.1 | Y | - |
| | 104.10 | 324.19 | 17.3% | 10.5% | 7.6% | 7.5% | 51.5% | 5.7% | 25.1% | 12.8% | 0.31 | 1.01 | 0.10 | 0.29 | -20.5 | Y | - |
| Interval 6 | 119.64 | 391.05 | 45.1% | 9.5% | 31.3% | 7.9% | 5.7% | 0.5% | 88.8% | 84.7% | 0.89 | 3.95 | 0.08 | 0.20 | - | - | 5.9 |
| | 119.75 | 391.79 | 19.2% | 8.0% | 53.8% | 17.5% | 1.5% | 0.0% | 93.0% | 97.4% | 0.98 | 3.08 | 0.00 | 0.15 | - | Y | - |
| | 121.66 | 403.66 | 52.0% | 11.6% | 28.2% | 7.4% | 0.8% | 0.1% | 98.6% | 97.4% | 0.98 | 3.81 | 0.09 | 0.92 | -24.3 | Y | 31.5 |
| | 123.43 | 414.42 | 39.9% | 9.9% | 38.7% | 9.4% | 1.8% | 0.2% | 95.7% | 95.5% | 0.97 | 4.11 | 0.10 | 0.42 | -22.2 | Y | - |
| | 123.82 | 416.86 | 92.5% | 3.6% | 3.4% | 0.2% | 0.3% | 0.1% | 99.7% | 92.6% | 0.96 | 16.85 | 0.18 | 0.47 | -22.1 | Y | - |
| | 124.02 | 418.06 | 65.1% | 12.1% | 21.6% | 0.7% | 0.4% | 0.1% | 99.5% | 98.4% | 0.99 | 29.62 | 0.23 | | -22.1 | Y | - |
| | 128.74 | 447.02 | 96.4% | 1.5% | 1.5% | 0.1% | 0.5% | 0.1% | 99.5% | 75.6% | 0.83 | 13.27 | 0.21 | 0.74 | - | - | 359.6 |
| | 129.77 | 453.32 | 83.8% | 14.6% | 1.1% | 0.1% | 0.3% | 0.1% | 99.6% | 77.3% | 0.97 | 15.71 | 0.22 | | -26.8 | - | - |
| | 129.84 | 453.73 | 83.8% | 14.7% | 1.1% | 0.1% | 0.3% | 0.1% | 99.7% | 81.6% | 0.98 | 8.54 | 0.20 | 0.15 | -25.2 | - | - |
| | 129.96 | 454.42 | 83.4% | 13.6% | 2.3% | 0.2% | 0.4% | 0.1% | 99.6% | 86.2% | 0.97 | 9.91 | 0.19 | | -27.3 | - | - |
| | 130.21 | 456.18 | 84.5% | 7.2% | 7.5% | 0.5% | 0.2% | 0.1% | 99.7% | 96.9% | 0.98 | 16.30 | 0.28 | | -28.2 | - | - |

**Table 1.** Table showing values in Magadi for GDGT relative abundances, index values, bulk geochemistry, and $C_{17:0}$ fatty acid abundances. Percent relative abundances of GDGT-0 to -4 and crenarchaeol (cren) and its crenarchaeol' (cren'). Additionally, indices related to methane cycling (i.e., % GDGT-0/Cren, % GDGT-2/Cren, and MI) and mesophilic archaeal inputs ([2] / [3]) are included as well as bulk $\delta^{13}C_{OM}$, Ca / Na, f[cren'], pyrite presence, and proportions of the $C_{17:0}$ fatty acid.