# Peer review of "Hot-spring inputs and climate drive dynamic shifts in archaeal communities in Lake Magadi, Kenya Rift Valley"

_EGUsphere, 2024_

## Referee Comment (RC1)

**Summary**

The main finding of this manuscript is there is abrupt microbial community shifts at Lake Magadi over the last 456 ky, alternating between periods with prominent methane cycling and periods without. The authors use multiple organic geochemical techniques, specifically isoGDGT indices, leaf waxes, and bulk organic matter d13C values, alongside previously published information on lake levels and hydrothermal inputs to make these claims. Intervals with strong methane cycling are associated with low hydrothermal inputs while intervals with weak methane cycling are associated with greater hydrothermal inputs.

**General comments**

Overall, the research findings are new and interesting, and most of the methodology is sound. Exploring biomarkers in soda lakes (and other such non-freshwater lakes) is certainly useful for testing biomarker applicability in a wider range of environments. However, the application of leaf waxes and the discussion would benefit from further development.

Here are my general comments:

1) Despite different lines of evidence being used, the discussion was very GDGT-reliant. Leaf wax data were included but the extent to which they were considered in the context of the paper was limited. For leaf waxes, the main measurements used were ACL and CPI. There was a heavy reliance of ACL and CPI as an indicator of terrestrial sourcing or C4 vegetation, but ACL and CPI alone are insufficient as determinations of either. Pollen records were cited (L 489-491) and would be useful for tracking C4 grassland abundance downcore alongside OM d13C (e.g., are C4 grasslands only abundant in interval 1?). If possible, a better metric of C4 vegetation would've been d13C of individual long-chain n-alkanes (e.g. C27, C29, and C31), rather than just bulk OM. Additionally, were there any patterns in changes of alkane or FAME abundances (both total and for individual chain lengths) downcore?

2) While the figures used as visuals do their job, modifications to current figures and additional figures would better support the main text and push discussion forward. For example, how do plots of CPI, ACL, and bulk OM d13C compare downcore? If C17 FAMEs are being used as an indicator for SRB, how do abundances compare downcore? What about C17 FAME abundance plotted alongside pyrite appearance and methane cycling indices? What about plotting [2]/[3] data from Rattanasriampaipong et al. (2022) alongside values from this study for a visual comparison of overlap? As for current figures, consider simply removing Fig. 2 as knowing the structure of the GDGTs being used don't contribute to an increase in understanding the findings of the manuscript, particularly since Fig. 2 is currently cited in Sect. 4.1 where visuals of GDGT structure are not very relevant. Fig. 3d shows OM d13C downcore, but the < -40 permil values in interval 2 makes it difficult to compare d13C values across intervals (it currently just looks like a straight line for every interval besides 2). One possible modification is to have the full d13C record as an inset graph and a larger graph excluding just the < -40 permil values. For Fig. 4, the link between hydrothermal inputs and MI, from the graph alone, is not immediately obvious. Based on information from the main text, increased hydrothermal input is indicated by low Ca/Na and high %REE. If low MI occurs when there's more hydrothermal input, I expected to see low Ca/Na and high %REE in the blue MI-off intervals, but this doesn't seem to be the case.

3) Is there more climate context for Lake Magadi over the study period? The African Humid Period was mentioned (and it needs to be cited in the main text) as occurring in interval 1, but were

there any climate events of note beyond interval 1 that could've contributed to our understanding of the biomarker records at Lake Magadi? Currently, the manuscript formulation implies much of the biomarker patterns observed are due to changes in hydrothermal inputs, but looking at Fig. 4, while hydrothermal inputs may explain some of the story, it doesn't seem to explain the whole story. If so, what are other drivers to the methane cycling indices?

**Specific comments**

Section 2.1: Were there any age estimates for paleolake shorelines? What about any lake level history records?

L 221-223: A more thorough explanation of the [2]/[3] index would be useful. What is this index an indicator of? Is it for distinguishing mesophilic from high/low MI environments?

Section 3.1, paragraph 1-2: The first few sentences of paragraph two are the same content as the Fig. 3 caption and can be deleted. I suggest then combining paragraph 1 and 2, specifying that the oscillations in GDGT indices correspond to shifts between intervals.

L 255-257: Were any samples taken and measured for biomarkers from presumed low-TOC sections? Is it possible samples presented here are not properly representative of the whole-core record due to selectively sampling only the dark, silty sections of the core?

Section 3.1, paragraph 3: Alongside the average index values, I suggest including the standard deviations.

Section 3.3: As mentioned, CPI is more a metric of degradation/diagenesis (something acknowledged later in the main text) rather than terrestrial sourcing. Broadly suggesting FAMEs to be terrestrially sourced (L 294) counters the point of the last paragraph in the section, which is that short-chain FAMEs are diagnostic of SRB in sediments and, in the context of the manuscript, presumably living in the lake. The evidence for SRB presence is also somewhat lacking. Four compounds were listed as possible indicators of SRB, C15:0, C15:0-iso, C17:0, and C17:0-iso FAMEs. It sounds like only 1 of the 4 compounds (C17:0 FAME) were identified in the 15 samples measured for FAMEs. Were there attempts to identify C15:0-iso and C17:0-iso in these samples? C17:0 FAMEs are not exclusively produced by SRB so the presence of these compounds cannot absolutely be attributed to SRB. Since pyrite presence did not always overlap with C17:0 FAME presence, are there other lines of evidence suggesting SRB presence?

L 347-352: The phrasing in these lines is confusing. It seems to imply there is AOM therefore we expect to see high methane cycling indices rather than the other way around. It also seems to imply AOM should exist because there is SRB which isn't always true.

L 387-388: This sentence should be moved earlier to the beginning of the paragraph as it does a much better job explaining how you know interval 2 is more influenced by methanogens than by AOM.

Section 4.1.2: The first half of the second paragraph repeats the info from the first paragraph (e.g., increase in crenarchaeol, low methane cycling indices) and should be consolidated to avoid redundancy.

Section 4.2: The majority of this section discusses hydrothermal input data cited in other papers with little mention of the links to this manuscript's findings until the very end. While interesting, it doesn't seem to merit a full section. The most relevant point is the last sentence so the rest of the section could reasonably be condensed and incorporated into another section (perhaps 4.1.2 MI-off periods).

Table 1: This table is more appropriate as an appendix rather than in the main text, or even excluded entirely and left as a submission to a data repository. I suggest condensing this table down to only feature average index values (along with their standard deviations) for each of the 6 intervals. Leave out the fractional GDGT abundances, pyrite presence, and C17:0 ng g^-1 sed extracted columns. Also, I suggest formatting the table using the table function in MS Word rather than copying directly from Excel. You may also consider turning some of the info from table 1 into a figure, perhaps a box-and-whisker plot showing each index for each interval.

Figure order: The figures are not ordered in the sequence they appear in the main text. The current order of figure appearance is 1, 3, 2, 5, 6, then 4, and should be renumbered in the order they appear.

**Technical corrections**

L 15-16: Since the biomarker data for this manuscript spans 456 ka to 15 ka, it's more accurate to say <500 ka rather than <700 ka.

L 29-30: References needed for "modern studies of both prokaryotic and eukaryotic organisms"

L 30-31: Extra space before period.

L 48: Replace "CH4" with "methane" for consistency.

L 67-68: This sentence can be condensed into the end of the previous paragraph by writing "Nitrososphaerota (formerly Thaumarchaeota) and Thermoproteota (formerly Crenarchaeota)" with their corresponding references.

L 71-73: Edit sentence for consistency. Something like "… representative of not only the Group 1 ANME consortium (ANME-1) that produce GDGTs, but also of Group 2 and Group 3 consortia (ANME-2 and ANME-3 respectively)."

L 76-78: References needed for "previous studies have used GDGT-0 and GDGT-2 ratioed to the GDGT crenarchaeol value…"

L 105: Duplicated the word "season". Should just be "rainy season".

L 117: Missing space before "Although Lake Magadi…"

L 125: Replace "partend" with "end". Replace semicolon separating the latitude and longitude with a comma.

L 128: Change "dated at ~ 1 Ma" to "dated to ~ 1 Ma"

L 130-132: Rearrange sentence from "… were subsampled and freeze-dried from dark brown to black silty clay intervals in the core" to "… were subsampled from dark brown to black silty clay intervals in the core then freeze-dried."

L 132: Replace "samples" with "intervals"

L 132-133: Specify reasons for high TOC assumption. Is it just the dark coloration of the sediment? Also, specify what is meant by "best results". Is it just higher yield of biomarkers?

L 135-137: Edit sentence for flow and consistency. From "… a large subhumid lake, when the freshwater lake was fed by rivers and groundwater continuously, to the small, tectonically restricted, saline alkaline pan partly fed by hot springs" to "… a large, freshwater, subhumid lake, fed continuously by rivers and groundwater, to a small, tectonically restricted, saline alkaline pan, partly fed by hot springs"

L 168: Change "step" to "rate"

L 201: Change "differ from" to "differ in"

L 201-202: Edit last part of sentence for clarification. Change "… even those that are saline and alkaline" to something like "particularly those in saline, alkaline environments."

L 210, 219, and 225: For consistency with eq. 1, use fractions when formatting eq. 2, 3, and 4. Change the "x" to a multiplication symbol and delete the percent symbol after 100.

L 226: Blank line labeled as eq. 5. Delete and renumber the equations that follow.

Formatting GDGT index names: Consider eliminating spaces within index names. For instance, writing "%GDGT-2/cren" instead of "% GDGT-2 / cren".

L 244-246: For depth values, use the same number of decimal places for consistency. I suggest using 2 decimal places for all values. Also, replace semicolons with commas.

L 250: In "cren'", change the apostrophe to the actual prime symbol. This should be the case for all subsequent instances of cren'.

L 274: Leave out mention of Interval 6 and just leave Interval 2 considering Interval 6 has not yet been discussed at this point in the results section.

L 279: Can remove first sentence and start paragraph with second sentence.

L 279-280: Specify what is meant by "similar pattern". Is it that bulk OM d13C oscillates between high and low values between intervals? If yes, say it.

L 341-342: Change "0.3 < MI < 0.5" to "MI < 0.5" as 0.3 doesn't seem to be a relevant value.

L 417: Replace "these intervals are" with "this interval is"

L 440: Remove "(Table 1)"

L 455-457: Change "green checkered patterns" to "blue regions". Also Fig. 3 was cited in parentheses twice.

Fig. 3 caption: Give the full names of the indices. Replace "%0/Cren" and "%2/Cren" with "%GDGT-0/Cren" and "%GDGT-2/Cren" respectively.

---

## Author Comment (AC1)

**Summary**

The main finding of this manuscript is there is abrupt microbial community shifts at Lake Magadi over the last 456 ky, alternating between periods with prominent methane cycling and periods without. The authors use multiple organic geochemical techniques, specifically isoGDGT indices, leaf waxes, and bulk organic matter d13C values, alongside previously published information on lake levels and hydrothermal inputs to make these claims. Intervals with strong methane cycling are associated with low hydrothermal inputs while intervals with weak methane cycling are associated with greater hydrothermal inputs.

**General comments**

Overall, the research findings are new and interesting, and most of the methodology is sound. Exploring biomarkers in soda lakes (and other such non-freshwater lakes) is certainly useful for testing biomarker applicability in a wider range of environments. However, the application of leaf waxes and the discussion would benefit from further development.

Here are my general comments:

1) Despite different lines of evidence being used, the discussion was very GDGT-reliant. Leaf wax data were included but the extent to which they were considered in the context of the paper was limited. For leaf waxes, the main measurements used were ACL and CPI. There was a heavy reliance of ACL and CPI as an indicator of terrestrial sourcing or C4 vegetation, but ACL and CPI alone are insufficient as determinations of either. Pollen records were cited (L 489-491) and would be useful for tracking C4 grassland abundance downcore alongside OM d13C (e.g., are C4 grasslands only abundant in interval 1?). If possible, a better metric of C4 vegetation would've been d13C of individual long-chain n-alkanes (e.g. C27, C29, and C31), rather than just bulk OM. Additionally, were there any patterns in changes of alkane or FAME abundances (both total and for individual chain lengths) downcore?

   - **After some consideration, we have decided to remove the *n*-alkanes from the paper as well as the FAMEs. This was, in part, due to the reviewers' comments stating that the extent of consideration was limited. While the *n*-alkanes provide some context to the core, they are not the main focus of the paper and will be removed.**

2) While the figures used as visuals do their job, modifications to current figures and additional figures would better support the main text and push discussion forward. For example, how do plots of CPI, ACL, and bulk OM d13C compare downcore? If C17 FAMEs are being used as an indicator for SRB, how do abundances compare downcore? What about C17 FAME abundance plotted alongside pyrite appearance and methane cycling indices?
   - **A plot was created to compare the CPI, ACL, and bulk 13C measurements. The C17 FAMEs have been removed from the paper after reconsidering the lack of overlap of C17 FAMEs and pyrite in the core.**

   What about plotting [2]/[3] data from Rattanasriampaipong et al. (2022) alongside values from this study for a visual comparison of overlap?
   - **More information has been placed in the text (Section 2.3.2) to better contextualize the results from the different samples and localities in Rattanasriampaipong et al. (2022) and how they relate to the research we presented in this paper.**

As for current figures, consider simply removing Fig. 2 as knowing the structure of the GDGTs being used don't contribute to an increase in understanding the findings of the manuscript, particularly since Fig. 2 is currently cited in Sect. 4.1 where visuals of GDGT structure are not very relevant.

- **After discussion with co-authors, Fig. 2 will be removed from the paper and simply cited as the reviewer notes.**

Fig. 3d shows OM d13C downcore, but the < -40 permil values in interval 2 makes it difficult to compare d13C values across intervals (it currently just looks like a straight line for every interval besides 2). One possible modification is to have the full d13C record as an inset graph and a larger graph excluding just the < -40 permil values.

- **The scale has been changed on the graph from a starting $\delta^{13}C$ of 0 to -15 ‰. A scale break has also been added between -35 and -40 ‰. The figure below shows the changes. The y-axis has also been scaled based on the scale break with the lower part scaled to 30% and the upper part scaled to 70%.**

[Figure]

For Fig. 4, the link between hydrothermal inputs and MI, from the graph alone, is not immediately obvious. Based on information from the main text, increased hydrothermal input is indicated by low Ca/Na and high %REE. If low MI occurs when there's more hydrothermal input, I expected to see low Ca/Na and high %REE in the blue MI-off intervals, but this doesn't seem to be the case.

- **We agree with the reviewer that it is not immediately obvious from Fig. 4 alone, this is why we decided to add a PCA plot and accompanying Spearman correlation matrix to help disentangle some of the complex relationships between REEs and methane indices. While Fig. 4 shows an overall trend of increased aridity and thus an increased proportion of hydrothermal flow to Lake Magadi steadily decreasing, but not a 1:1 relationship of Ca/Na from Interval 5 to Interval 1 (which is discussed further in the next general comment below), it does not detail the relationship between REEs as clearly which is why we chose to perform a PCA and Spearman correlation. The PCA loads the methane indices and REEs and shows an anti-correlation between these two groups. This anti-correlation can also be seen in the Spearman correlation matrix. So, while Fig. 4 does not show these trends as clearly, we hope that the PCA at least clarifies the relationship between REEs and methane indices.**
- **We have also clarified this by adding Ca/Na to the PCA and correlation matrix (shown below) further solidifying the relationships between methane indices and**

proportionally more hydrothermal inputs to Lake Magadi. Tb has been removed from the REEs as it had a very low number of samples where it was found (n=4) and was thus skewing interpretations in the core since other REEs and indices were represented more robustly (n≥32). The Ca/Na is anti-correlated with REEs in both the PCA and correlation matrix. Since the proportion of Ca/Na decreases when REEs increase, we can say that statistically, when it is drier (and thus proportionately more hydrothermally influenced) the Ca/Na decreases, REE values increase, and the methane indices are suppressed.

• Ca/Na is loaded positively on PC1 and PC2, while the methane indices ([2]/[3], MI, %0/Cren, and %2/Cren) are loaded positively on PC1 and negatively on PC2. This indicates these two measurements are statistically different from one another. So, while the values in Fig. 4 don't appear to have the 1:1 relationship that would be expected with the proportional increase in hot spring activity, statistically these two are different enough from one another.

[Figure]

3) Is there more climate context for Lake Magadi over the study period? The African Humid Period was mentioned (and it needs to be cited in the main text) as occurring in interval 1, but were there any climate events of note beyond interval 1 that could've contributed to our understanding of the biomarker records at Lake Magadi? Currently, the manuscript formulation implies much of the biomarker patterns observed are due to changes in hydrothermal inputs, but looking at Fig. 4, while hydrothermal inputs may explain some of the story, it doesn't seem to explain the whole story. If so, what are other drivers to the methane cycling indices?

- **We recognize that we did not fully describe the climatic context of this work and note that the following three papers discuss the climate in this region of the East African Rift Valley in more detail:**
    - **Owen, R. B., Muiruri, V. M., Lowenstein, T. K., Renaut, R. W., Rabideaux, N., Luo, S., ... & Mbuthia, A. (2018). Progressive aridification in East Africa over the last half million years and implications for human evolution. Proceedings of the National Academy of Sciences, 115(44), 11174-11179.**
    - **Owen, R. B., Renaut, R. W., Muiruri, V. M., Rabideaux, N. M., Lowenstein, T. K., McNulty, E. P., ... & Stockhecke, M. (2019). Quaternary history of the Lake Magadi Basin, southern Kenya Rift: Tectonic and climatic controls. Palaeogeography, palaeoclimatology, palaeoecology, 518, 97-118.**
    - **Muiruri, V. M., Owen, R. B., Lowenstein, T. K., Renaut, R. W., Marchant, R., Rucina, S. M., ... & Wang, C. (2021). A million year vegetation history and palaeoenvironmental record from the Lake Magadi Basin, Kenya Rift Valley. Palaeogeography, Palaeoclimatology, Palaeoecology, 567, 110247.**
- **Briefly, in the region around Lake Magadi in the time ranges of this study (ca. 456 to 14.9 ka) there was a gradual aridification noted by _proportionally_ larger inputs of hydrothermal flow, rather than an increase in hydrothermal activity, compared to other meteoric sources (i.e. rainfall and outwash from riverine inputs). That is, during drier periods when there was less rainfall, the higher proportions of hydrothermal flow maintained a perennially wet lake and there is not necessarily more activity from hydrothermal springs, we are seeing more evidence of the springs from a reduced dilution factor of freshwater inputs. Additionally, there are no noted mud cracks, erosion, or other evidence of the lakebed completely drying. These proportions are noted by increases in rare earth elements (REEs) and a gradual decrease in the proportion of Ca/Na. A noted exception is Interval 5, which we will reassess as there is a large spike in the Ca/Na and a dearth of REEs, lower than even intervals where proportions of hydrothermal input should be lower. One possible explanation is that Interval 5 is actually much wetter and more well-mixed such that an increase in other meteoric sources are having an impact on both the lake and the archaeal communities.**
    - **Additionally, after discussing it with other co-authors, the tephra seen in Interval 5 is likely much too altered (zeolitized) and no longer considered to be tephra. This is because tephra tends to rapidly alter under the conditions in Lake Magadi (i.e., high pH).**
    - **The notation of tephra will be changed on Figs. 3 and 4 to that of a perennial lake rather than as tephra ashfall.**

**Specific comments**

Section 2.1: Were there any age estimates for paleolake shorelines? What about any lake level history records?

- **Ages have been added from Renaut and Owen (2023).**

L 221-223: A more thorough explanation of the [2]/[3] index would be useful. What is this index an indicator of? Is it for distinguishing mesophilic from high/low MI environments?

- **We have taken this into account and added more information about the [2]/[3] index from Rattanasriampaipong et al. (2022)**

Section 3.1, paragraph 1-2: The first few sentences of paragraph two are the same content as the Fig. 3 caption and can be deleted. I suggest then combining paragraph 1 and 2, specifying that the oscillations in GDGT indices correspond to shifts between intervals.

- **The redundant text has been removed and the paragraphs have been combined for clarity as suggested**

L 255-257: Were any samples taken and measured for biomarkers from presumed low-TOC sections? Is it possible samples presented here are not properly representative of the whole-core record due to selectively sampling only the dark, silty sections of the core?

- **One reason that the strategy appears to not be optimal is a result of the relatively poor recovery of the core at ca. 55.4% (Line 126). As for the large spatial differences of 3-15 m between single intervals these are a result of areas with poorer core recovery where there was either no sample or the skipped intervals were too mineral rich or simply a brecciated material that could not be effectively sampled.**

Section 3.1, paragraph 3: Alongside the average index values, I suggest including the standard deviations.

- **These have been added to the manuscript text.**

Section 3.3: As mentioned, CPI is more a metric of degradation/diagenesis (something acknowledged later in the main text) rather than terrestrial sourcing. Broadly suggesting FAMEs to be terrestrially sourced (L 294) counters the point of the last paragraph in the section, which is that short-chain FAMEs are diagnostic of SRB in sediments and, in the context of the manuscript, presumably living in the lake. The evidence for SRB presence is also somewhat lacking. Four compounds were listed as possible indicators of SRB, C15:0, C15:0-iso, C17:0, and C17:0-iso FAMEs. It sounds like only 1 of the 4 compounds (C17:0 FAME) were identified in the 15 samples measured for FAMEs. Were there attempts to identify C15:0-iso and C17:0-iso in these samples? C17:0 FAMEs are not exclusively produced by SRB so the presence of these compounds cannot absolutely be attributed to SRB. Since pyrite presence did not always overlap with C17:0 FAME presence, are there other lines of evidence suggesting SRB presence?

- **The FAMEs and their interpretations have been removed from the manuscript.**

L 347-352: The phrasing in these lines is confusing. It seems to imply there is AOM therefore we expect to see high methane cycling indices rather than the other way around. It also seems to imply AOM should exist because there is SRB which isn't always true.

- **This has been removed and modified based on the suggestions of both reviewers regarding the FAMEs and their interpretations in the core.**

L 387-388: This sentence should be moved earlier to the beginning of the paragraph as it does a much better job explaining how you know interval 2 is more influenced by methanogens than by AOM.

- **This has been rearranged**

Section 4.1.2: The first half of the second paragraph repeats the info from the first paragraph (e.g., increase in crenarchaeol, low methane cycling indices) and should be consolidated to avoid redundancy.

- **This has been consolidated and the repeated text has been removed**

Section 4.2: The majority of this section discusses hydrothermal input data cited in other papers with little mention of the links to this manuscript's findings until the very end. While interesting, it doesn't seem to merit a full section. The most relevant point is the last sentence so the rest of the section could reasonably be condensed and incorporated into another section (perhaps 4.1.2 MI-off periods).

- **We have clarified and condensed parts of this section per the reviewer's request, but we do feel that this section is important and distinct enough to remain separate from 4.1.**

Table 1: This table is more appropriate as an appendix rather than in the main text, or even excluded entirely and left as a submission to a data repository. I suggest condensing this table down to only feature average index values (along with their standard deviations) for each of the 6 intervals. Leave out the fractional GDGT abundances, pyrite presence, and C17:0 ng g^-1 sed extracted columns. Also, I suggest formatting the table using the table function in MS Word rather than copying directly from Excel. You may also consider turning some of the info from table 1 into a figure, perhaps a box-and-whisker plot showing each index for each interval.

- **The table has been shrunk per the author's suggestions and the more detailed remaining data will be uploaded as a supplementary table to Biogeosciences.**

Figure order: The figures are not ordered in the sequence they appear in the main text. The current order of figure appearance is 1, 3, 2, 5, 6, then 4, and should be renumbered in the order they appear.

- **The Figure numbers have been corrected. The figure which had GDGT structures has been removed (Formerly Fig. 2).**

**Technical corrections**

*These will be addressed once editor approval has been granted for submitting the manuscript.*

L 15-16: Since the biomarker data for this manuscript spans 456 ka to 15 ka, it's more accurate to say <500 ka rather than <700 ka.

L 29-30: References needed for "modern studies of both prokaryotic and eukaryotic organisms"

L 30-31: Extra space before period.

L 48: Replace "CH4" with "methane" for consistency.

L 67-68: This sentence can be condensed into the end of the previous paragraph by writing "Nitrososphaerota (formerly Thaumarchaeota) and Thermoproteota (formerly Crenarchaeota)" with their corresponding references.

L 71-73: Edit sentence for consistency. Something like "… representative of not only the Group 1 ANME consortium (ANME-1) that produce GDGTs, but also of Group 2 and Group 3 consortia (ANME-2 and ANME-3 respectively)."

L 76-78: References needed for "previous studies have used GDGT-0 and GDGT-2 ratioed to the GDGT crenarchaeol value…"

L 105: Duplicated the word "season". Should just be "rainy season".

L 117: Missing space before "Although Lake Magadi…"

L 125: Replace "partend" with "end". Replace semicolon separating the latitude and longitude with a comma.

L 128: Change "dated at ~ 1 Ma" to "dated to ~ 1 Ma"

L 130-132: Rearrange sentence from "… were subsampled and freeze-dried from dark brown to black silty clay intervals in the core" to "… were subsampled from dark brown to black silty clay intervals in the core

then freeze-dried."

L 132: Replace "samples" with "intervals"

L 132-133: Specify reasons for high TOC assumption. Is it just the dark coloration of the sediment? Also, specify what is meant by "best results". Is it just higher yield of biomarkers?

L 135-137: Edit sentence for flow and consistency. From "… a large subhumid lake, when the freshwater lake was fed by rivers and groundwater continuously, to the small, tectonically restricted, saline alkaline pan partly fed by hot springs" to "… a large, freshwater, subhumid lake, fed continuously by rivers and groundwater, to a small, tectonically restricted, saline alkaline pan, partly fed by hot springs"

L 168: Change "step" to "rate"

L 201: Change "differ from" to "differ in"

L 201-202: Edit last part of sentence for clarification. Change "… even those that are saline and alkaline" to something like "particularly those in saline, alkaline environments."

L 210, 219, and 225: For consistency with eq. 1, use fractions when formatting eq. 2, 3, and 4. Change the "x" to a multiplication symbol and delete the percent symbol after 100.

L 226: Blank line labeled as eq. 5. Delete and renumber the equations that follow.

Formatting GDGT index names: Consider eliminating spaces within index names. For instance, writing "%GDGT-2/cren" instead of "% GDGT-2 / cren".

L 244-246: For depth values, use the same number of decimal places for consistency. I suggest using 2 decimal places for all values. Also, replace semicolons with commas.

L 250: In "cren'", change the apostrophe to the actual prime symbol. This should be the case for all subsequent instances of cren'.

L 274: Leave out mention of Interval 6 and just leave Interval 2 considering Interval 6 has not yet been discussed at this point in the results section.

L 279: Can remove first sentence and start paragraph with second sentence.

L 279-280: Specify what is meant by "similar pattern". Is it that bulk OM d13C oscillates between high and low values between intervals? If yes, say it.

L 341-342: Change "$0.3 < MI < 0.5$" to "$MI < 0.5$" as 0.3 doesn't seem to be a relevant value.

L 417: Replace "these intervals are" with "this interval is"

L 440: Remove "(Table 1)"

L 455-457: Change "green checkered patterns" to "blue regions". Also Fig. 3 was cited in parentheses twice.

Fig. 3 caption: Give the full names of the indices. Replace "%0/Cren" and "%2/Cren" with "%GDGT- 0/Cren" and "%GDGT-2/Cren" respectively.

---

## Author Comment (AC2)

The authors investigated sections of a drill core from Lake Magadi (Hominin Sites and Paleolakes Drilling Project, HSPDP), a soda lake in Kenya, to reconstruct the microbial methane cycle of the lake system over the last 456 ka. The study is focused on molecular biomarker analysis, especially isoGDGTs, representing archaeal core lipids. Together with accompanying (organic-) geochemical data and published information, the authors interpret periodical shifts in microbial methane cycling (and consequently the archaeal community) to be associated with changes in the hydrothermal input at Lake Magadi. It is indicated that phases of low hydrothermal activity show increased microbial methane cycling as compared to phases of high hydrothermal activity.

Soda lakes are important habitats for life. Their investigation, including the microbial methane cycle over time, provides valuable information on these extreme environments and potential early Earth habitats. A detailed reconstruction of the microbial methane cycle of Lake Magadi over time does not exist so far. The findings of the study by Collins et al. are new, complement existing data, and improve our understanding of the Magadi system. The used core samples from the HSPDP are unique and represent excellent material to study archaeal communities/the microbial methane cycle of Lake Magadi over time. However, the manuscript needs to be substantially improved in some areas before publication:

1) The sampling strategy is not optimal (cf., l. 129–133). The authors focused on samples that were expected to have high total organic carbon contents (data not presented in the manuscript), which was only assessed by visual inspection (dark brown to black silty clay). The authors argue that those samples would yield the best results. This may have created a biased data set (also samples with low organic carbon contents may show a great molecular diversity). Additionally, the sampling scheme is not consistent. Between the defined intervals #1–6, several meters of core are not covered (3–15 m between single intervals), while within an interval the sampling steps are in parts as close as a few centimeters. It would be interesting to see, if microbial methane cycling was also active during the deposition of sediments with low organic carbon content.

   - **The reason that the strategy appears to not be optimal is in part a result of the relatively poor recovery of the core at ca. 55.4% (Line 126). As for the large spatial differences of 3-15 m between single intervals these are a result of areas with poorer core recovery where there was either no sample or the skipped intervals were too mineral rich or simply a brecciated material that could not be effectively sampled. We agree that a more ideal sampling strategy would be better, but it was not possible with the core that we have available.**

2) The study lacks bulk geochemical data of the samples, which would be important to contextualize the presented biomarker and isotope data (e.g., total organic and inorganic carbon contents, total sulfur content, bulk $^{13}C_{carb}$). Especially stable carbon isotope data of the carbonate phase ($\delta^{13}C_{carb}$) would improve the discussion of shifts in methane cycling (it seems that at least some samples contain carbonate, as the samples were acid-leached before $^{13}C_{org}$ analysis; l. 174–175).

   - **We agree with the reviewer, but the grant which was funding this research is no longer funded so we cannot go back and measure the total sulfur or the bulk $^{13}C_{carb}$. We have %TOC data as $LOI_{500}$ and will include these data, but due to concerns of high temperatures (ca. 1000 C) potentially combusting the Na carbonates in the samples**

**and creating lime in the furnace thus leading to potential fires, the %TIC values were not collected.**

3) The presented bulk $\delta^{13}C_{org}$ data lack context. In lake systems primary production and/or terrestrial input usually govern the carbon cycle. The presented data do not allow the assessment of the role of microbial methane cycling in the lake's carbon cycle over time. It would help to present total abundances of compounds in relation to the total organic carbon content (amount per g TOC). In addition, the $^{13}C_{org}$ data should be discussed together with the leaf wax data to evaluate the influence of terrestrial input on the $^{13}C_{org}$ values. In the presented data set, only three values in interval #2 indicate methanotrophy (−48.1‰, −64.2‰, −89.4‰; Table 1), the rest of the $^{13}C_{org}$ values could also be explained by variations in primary production and/or terrestrial input.

- **We will add a brief discussion of the major factors affecting d13org in lakes, noting that there is very little terrestrial input to Magadi through much of the record (as noted by n-alkane abundances), so that in lake processes likely dominate the overall C isotope systematics. Among in lake factors are included primary production, but also significant microbial primary and secondary production.**

4) The discussion of microbial sulfate reduction in the system (e.g., l. 366–379) is not based on a solid data set. In the current version, only $C_{17:0}$ FAME and the appearance of pyrite are used to track microbial sulfate reduction. The $C_{17:0}$ FAME, however, is not only produced by sulfate reducers and represents a weak biomarker. Furthermore, it seems that only few samples contain $C_{17:0}$ FAME, and it does not necessarily co-occur with pyrite (cf., Table 1). The authors also speculate on the sulfate availability without presenting any robust indication on sulfate levels. Without further data (e.g., sulfur content, stable sulfur isotope composition of the pyrites) this part of the discussion needs to be significantly reduced.

- **We agree that this is overly speculative, so we plan to remove this part of the discussion from the paper. While there is a possible connection in the intervals where $C_{17:0}$ FAME and pyrite overlap, the lack of data outlined above (e.g., sulfur content, stable sulfur isotope composition of the pyrites) means that this connection to sulfate reduction is a speculative one.**

5) The interpretation of increased microbial methane cycling at times of low hydrothermal input (and vice versa) is mainly based on the correlation of MI with REE data and Ca/Na-ratio. These data sets, however, do not always match (cf., figure 4). The authors should discuss the discrepancies in more detail, and present some explanations for the major discrepancies (e.g., low Ca/Na at the end of interval #2, high Ca/Na together with low REE abundance in interval #5, high REE abundance together with low Ca/Na at the end of interval #6). The MI data set seems to be much more consistent.

- **See answer to Reviewer #1's General Comment #3.**

**More specific comments:**

- The title is misleading, as the manuscript is focusing on the reconstruction of the microbial methane cycle in Lake Magadi over time, driven by archaea, and not on the reconstruction of the entire microbial community and its change over time. Please replace "microbial" in the title by "archaeal".

- o **This has been changed.**

- The errors for the $\delta^{13}C_{org}$ analyses should be presented (results section and Table 1).

    - o **These values have been added**

- I suggest including more details on the statistical evaluation (Fig. 5; PCA and correlation matrix) into the methods section.

    - o **A new section has been added in "Materials and Methods" (2.4) to address these deficiencies.**

- Why do the authors think the fatty acids $<C_{16:0}$ are degraded in the samples? I do not see any indication why this should be the case. The compounds were likely never present or below detection limit.

    - o **We agree with the reviewer and have changed the manuscript to reflect this change as there is no way for us to determine whether the FAMEs were degraded, ever produced, or simply below instrument detection limits. Additionally, *n*-alkanes and FAMEs have been removed from the manuscript. See reasoning in answers to Reviewer #1.**

- In section 4.1.1 the authors discuss missing pyrite in some intervals and explain this by too small pyrite aggregates that could not be seen by the naked eye and/or sulfur incorporation into kerogen (l. 406–408). This is pure speculation. The authors could have easily checked the samples for small pyrite aggregates by using thin section microscopy and could have measured the total sulfur content.

    - o **We agree, however, when the core was initially being described there were other items that were prioritized and now we do not have the funds to reevaluate core sections with thin section microscopy**

- The headline of section 4.2 should be changed to something like "The influence of hydrothermal activity on the microbial methane cycle".

    - o **We agree and have made a change to reflect this suggestion.**

- The REE data should be discussed in more detail in section 4.2.

    - o **We have expanded on the REEs in the text to better contextualize the hydrothermal inputs and how these REEs relate to those inputs.**

- Figures 3 and 4 should be turned 90° and stretched (differences e.g. in interval #1 are barely visible in the current version), with age/depth on the y-axis. It would also be important to include the stratigraphic units and different lithologies.

    - o **We appreciate that the reviewers have preferences for figure orientation, but we find that the information is well-conveyed as the figures are currently. If editors insist, we can make the suggested change, but feel it is not necessary to the paper.**

- Please carefully check the color coding of the symbols in figure 6. Shouldn't the cross at ca. 67% crenarchaeol be green or is the cross incorrect? What about the triangle at ca. 6% GDGT-2 (maybe blue or incorrect symbol)?

    - **We have reviewed the values to make sure there were no errors in how we reported the data and there is one anomalous value in Interval 3 as discussed on Line 269. Otherwise, all of the data appear to be correct on the ternary plot.**

- Please add some representative GC chromatograms for each interval to the supplement.

    - **Since we have removed the *n*-alkanes and FAMEs, LC chromatograms of the GDGTs have been added; see below for representative chromatograms from each Interval outlined in the manuscript:**

[Figure]

**Minor comments:**

*These will be addressed once editor approval has been granted for submitting the manuscript.*

L. 30: Please list some studies that have investigated soda lake sediments/sedimentary rocks over geologic time scales here (some are already mentioned in the manuscript, incl. those from Lake Magadi).

L. 30/31: Delete space before full stop.

L. 87: Delete bracket in front of [2].

L. 92: Replace "microbial" by "archaeal" (please also do so in other relevant areas of the manuscript not mentioned here).

L. 95: The "n" of *n*-alkanes should be written in italics.

L. 117: Insert space in front of "Although"; delete comma behind "it".

L. 188–190: Please check for correct phrasing (verb missing?).

L. 282: Please also calculate a mean value without the three outliers.

L. 324–326: Please check for correct phrasing (verb missing?).

L. 336: Replace "microbial" by "archaeal".

L. 349: Change "biomarkers" to "a potential biomarker".

L. 350: Change "(FAMEs) were identified" to "($C_{17:0}$ FAME) was identified".

L. 456: What do the authors mean by the "green checkered pattern"?

L. 606: Replace "predominantly microbial inputs" by "archaeal communities".

L. 607: Delete "archaeal".

Figure caption of figure 6: High MI is shown in green, not yellow.

Table 1: It would be great, if the color for "MI on" periods in the table would match the color used in the figures (green).

**Citation**: https://doi.org/10.5194/egusphere-2024-3006-RC2

---

## Referee Report (RR1)

**Summary**

The main finding of this manuscript is that the Lake Magadi archaeal community has abruptly shifted between periods with prominent methane cycling and periods without over the last ~500 ky. The authors use organic geochemical measurements, isoGDGT indices and bulk organic matter d13C values, as indicators of methane cycling, with evidence for both methanotrophy and methanogenesis. Periods with strong methane cycling were associated proportionally lower hydrothermal inputs, indicated by lower REE abundance.

**General comments**

The research findings are new and interesting, and the methodology is sound. As previously stated, exploring unusual lakes like soda lakes and providing a context for organic geochemical applications is useful for future work on other unusual lakes (which there are many of in the world). I don't believe there are any major/pressing problems in the current version of the manuscript as they've largely been corrected during the first round of revisions. However, there are some smaller issues that should be resolved prior to publication. See below.

**Specific comments**

L 80-81: The optimum temperature of what?

Sect 2.1, paragraph 2-3: For the formations/beds named, is it known how deep the lake was when these formations/beds were deposited? Or more specifically, is the presence of a formation at certain place evidence of a specific lake depth? The phrasing of paragraph 2 implies that the formations/beds were deposited at a specific time at some point within very large age ranges (e.g., "Oloronga Beds in outcrop have been dated between ~800 and 300 ka" sounds like the beds were deposited sometime within this 500 ky range which is very large). Is this actually the case or does it actually mean the formation starts/ends in this range? If it's the former, then such large age range estimates are not very useful for any kind of chronological determinations. If it's the latter, the phrasing of paragraph 2 needs to be changed (e.g., "Oloronga Beds in outcrop spans ~800-300 ka" or "Oloronga Beds in outcrop were lain down from ~800 to 300 ka"). Later in sect 4.1.1, L 413-414 states, "Earlier water depths are also unclear because accommodation space was changing as the axial rift developed with faulting and subsidence." Is this true for most of the time period within which these formations/beds were lain down? If so, then it merits explaining early on (e.g., somewhere in sect 2.1) why past determinations of lake depth is so difficult.

Sect 2.1, L 116-120: The part starting from "More recently…" to "… High Magadi Formation" is not super relevant and can likely be condensed (into the previous paragraph). Discussing renaming/reassignment of beds/formations without giving additional context makes it confusing as there were multiple new names (e.g., "Evaporite Series") not mentioned in the previous paragraph."

Sect 2.1, end of paragraph 4: Per responses to the first round of reviews, the manuscript may benefit from explicitly stating that the reason for the non-ideal sampling strategy is due to poor core recovery/poor core quality. This can be done at author's discretion.

Sect 2.2: Remove the 2.2 heading "Leaf wax and bulk organic preparation and analysis. Now that the leaf wax protocol is gone, sections 2.2.1 and 2.2.2 can probably be their own sections, that is, 2.2 and 2.3. Subsequently, the following subsections will have to be renumbered.

Sect 2.4: REE was defined as rare earth elements during the second instance it's mentioned (L 226) and not the first (L 222).

Sect 2: Since pyrite evidence is mentioned frequently, the methods for pyrite measurement/detection should be included somewhere in section 2.

Sect 4.1.1, paragraph 1 and 2: Suspected SMTZ presence is mentioned for the first time in paragraph 1, but evidence for a potential SMTZ at Lake Magadi is not presented until the next paragraph. L 317-319 states where SMTZ is suspected, we expect increases in methane GDGT indices, but L 340-342 states high methane GDGT indices contributes to evidence for SMTZ presence. The way this is written has some circular reasoning. The latter (using biomarker and geochemical evidence to make a statement) is more logical than the former (using an assumption to suggest expected patterns in biomarker and geochemical evidence).

L 385: If meromixis is a present characteristic of Lake Magadi, should explicitly state this fact in the sect 2.1 study location description.

Sect 4.1.2, paragraph 3: Both Ca/Na and REEs are mentioned here. It may merit briefly reminding/explaining what each represent (e.g., REEs come from hydrothermal inputs) as neither was explicitly defined in the context of Lake Magadi prior to this section.

Sect 4.1.2, last paragraph: Delete "The CPI$_{alk}$ and CPI$_{FA}$ 510 averages were 4.6 and 5.0, indicating more terrestrial input." Consider also deleting the following sentence, "So, while these values are lower than Interval 3, and closer to the values in Interval 1, these still indicate a higher terrestrial input during this timeframe," as now there's no longer a discussion of terrestrial inputs. If this is deleted, then the only sentence left in the paragraph (starting with "Lastly, …") would better be moved to the end of the previous paragraph to prevent having an orphan sentence.

**Technical corrections**

L 17-18: Replace "MI "switches off" (MI < 0.2); and on (MI > 0.5)" with "The MI switches "off" (MI < 0.2) and "on" (MI > 0.5)"

L 43: Change "generation of" to "the"

L 69: Remove "Moreover"

L 110: Remove "More recently" as it is used to start the next paragraph.

L 120: Remove "Various"

L 121-122: This line, "Although Lake Magadi is situated near the equator, it lies in a rain shadow. Consequently, today it has a large moisture deficit (2400 mm evaporation versus 500 mm precipitation annually: Damnati and Taieb, 1995)" would better fit as the last line of sect 2.1, paragraph 1, rather than where it is currently at.

L 211: Change "was used here" to "is used here"

Equation 4: Un-bold the text

L 222: Change "intervals of focus (i.e., Intervals 1, 3, and 5)" to "intervals of focus in the sediment core (designated as intervals 1, 3, and 5 in the results and discussion sections)"

L 227-231: Condense these lines, starting from "Prior to performing...", to "The data was tested for normality via the built-in "Normality and Lognormality Tests" function in GraphPad$^{©}$. Tests yielded lognormal distributions of each dataset and found the data to be non-normally distributed."

L 246: Is "197 to 149 ka" supposed to have a "ca." before it like the other intervals?

L 249: Remove "Table S1Table S1"

L 262: Change "46.6815 m" to "46.68 m"

L 263: Remove spaces in "%GDGT-0 / cren"

L 277: Remove "Table S1" from the start of the sentences.

L 291: Change "namely" to "such as"

L 344: This seems to refer to Figure 5 rather Figure 4.

L 345-346: This seems to refer to Figure 2 and 5 rather than 3 and 5.

L 346-347: Change "resulting from a more prevalent" to "as seen in a high".

L 354: Change "reported values" to "reported $\delta^{13}C_{OM}$ values" (assuming Summons et al. (1998) reported the $\delta^{13}C$ for OM)

L 402: Put Table S1 in parenthesis.

L 406-407: Split up the sentence at "and while", ending the first with "of the MI" and starting the next with "while both".

L 420: Flip "(MI-on; Fig. 2)" to "(Fig. 2; MI-on)" for consistency with the earlier parathesis.

L 424: Change "such as hot spring mats in Thermoproteota" to "such as in hot spring mats made by Thermoproteota".

L 446-449: Shorten the two sentences (as they're currently a bit redundant) to something like "In agreement with the pollen record, the $\delta^{13}C_{OM}$ values likely record a mixture of C4 grasses and C4 sedges. Similar $\delta^{13}C$ values were reported in $C_{27}$ to $C_{33}$ n-alkanes in equatorial regions of Cameroon, ranging from -18.2 to -17.6 ‰ and recording the signals from C4 grasses and sedges (Garcin et al., 2014)."

L 497: Can delete "(averaging 1.5 excluding the outlying value of 36.7). This average is closer to shallow Group I.1a Thermoproteota as described previously" as much of this was already stated in an earlier sentence.

L 526-527: Add a comma between "Magadi ash" so it's "Magadi, ash".

L 545: Remove spaces in "Ca / Na" so it becomes "Ca/Na".

L 551: This seems to refer to Fig. 4a rather than 5b.

L 558: Change "spring-runoff ratios" to "spring/runoff ratios" for consistency.

L 580: Should replace "RBO" with the full name of the co-author.

L 581: Change "special thank" to "special thanks".

L 582: Change "for providing, who provided" to "for providing".

L 584: Change "CSDF" to "CSD".

L 586: I assume "#XXX" will be replaced with an appropriate publication number once this paper is officially published?

Fig 2d: There seems to be a layer order issue with the data points in intervals 2, 4, and 6 plotting behind the yellow box instead of in front.

L 983-984: Remove the spaces between all the ratio indices (i.e., "%0/Cren" instead of "% 0 / Cren").

L 985-986: Remove "Checkered patterns indicate periods of tuffaceous silt deposit, which align with the low MI intervals" as there is no longer a checkered pattern in the figure.

L 991: Add "and in the Y-axis scale for $\delta^{13}C_{OM}$" after "X-axis scale".

L 995-996: Remove "The checkered pattern is indicative of periods of higher inferred hydrothermal flow" as there is no longer a checkered pattern in the figure.

---

## Author Response (AR2)

**Referee #1**

**Summary**

The main finding of this manuscript is that the Lake Magadi archaeal community has abruptly shifted between periods with prominent methane cycling and periods without over the last ~500 ky. The authors use organic geochemical measurements, isoGDGT indices and bulk organic matter d13C values, as indicators of methane cycling, with evidence for both methanotrophy and methanogenesis. Periods with strong methane cycling were associated proportionally lower hydrothermal inputs, indicated by lower REE abundance.

**General comments**

The research findings are new and interesting, and the methodology is sound. As previously stated, exploring unusual lakes like soda lakes and providing a context for organic geochemical applications is useful for future work on other unusual lakes (which there are many of in the world). I don't believe there are any major/pressing problems in the current version of the manuscript as they've largely been corrected during the first round of revisions. However, there are some smaller issues that should be resolved prior to publication. See below.

**Specific comments**

L 80-81: The optimum temperature of what?

**Author Comment: This is supposed to read "However, the optimum temperature for crenarchaeol production is closer to 40-45 oC (Zhang et al., 2006)." This has been changed and rearranged on lines 78-79 for clarity.**

Sect 2.1, paragraph 2-3: For the formations/beds named, is it known how deep the lake was when these formations/beds were deposited? Or more specifically, is the presence of a formation at certain place evidence of a specific lake depth? The phrasing of paragraph 2 implies that the formations/beds were deposited at a specific time at some point within very large age ranges (e.g., "Oloronga Beds in outcrop have been dated between ~800 and 300 ka" sounds like the beds were deposited sometime within this 500 ky range which is very large). Is this actually the case or does it actually mean the formation starts/ends in this range? If it's the former, then such large age range estimates are not very useful for any kind of chronological determinations. If it's the latter, the phrasing of paragraph 2 needs to be changed (e.g., "Oloronga Beds in outcrop spans ~800-300 ka" or "Oloronga Beds in outcrop were lain down from ~800 to 300 ka"). Later in sect 4.1.1, L 413-414 states, "Earlier water depths are also unclear because accommodation space was changing as the axial rift developed with faulting and subsidence." Is this true for most of the time period within which these formations/beds were lain down? If so, then it merits explaining early on (e.g., somewhere in sect 2.1) why past determinations of lake depth is so difficult.

**Author Comment: The clarity has been added by saying that "The chert-bearing Oloronga Beds in outcrop have been dated were lain down between ~ 800 and 300 ka..." for clarity. This is only for the early formation of the lake and not for our samples collected where the lakebed has been relatively tectonically stable from ca. 540 ka to present.**

Sect 2.1, L 116-120: The part starting from "More recently..." to "... High Magadi Formation" is not super relevant and can likely be condensed (into the previous paragraph). Discussing renaming/reassignment of beds/formations without giving additional context makes it confusing as there were multiple new names (e.g., "Evaporite Series") not mentioned in the previous paragraph."

**Author Comment: This text has been removed and only the text which reads "Although Lake Magadi is situated near the equator, it lies in a rain shadow. Consequently, today it has a large moisture deficit (2400 mm evaporation versus 500 mm precipitation annually: Damnati and Taieb, 1995)." Was kept and was moved to the end of paragraph 1 of Section 2.1 per the reviewer's suggestion.**

Sect 2.1, end of paragraph 4: Per responses to the first round of reviews, the manuscript may benefit from explicitly stating that the reason for the non-ideal sampling strategy is due to poor core recovery/poor core quality. This can be done at author's discretion.

**Author Comment: This has been added on L 131.**

Sect 2.2: Remove the 2.2 heading "Leaf wax and bulk organic preparation and analysis. Now that the leaf wax protocol is gone, sections 2.2.1 and 2.2.2 can probably be their own sections, that is, 2.2 and 2.3. Subsequently, the following subsections will have to be renumbered.

**Author Comment: We believe that the sections are sufficient to stay as-is per the suggestions of Referee #2 who suggested that we change the section title from "Leaf wax and bulk organic preparation and analysis" to "GDGT and bulk organic preparation and analysis".**

Sect 2.4: REE was defined as rare earth elements during the second instance it's mentioned (L 226) and not the first (L 222).

**Author Comment: We have changed this discrepancy on L 217 and L 221.**

Sect 2: Since pyrite evidence is mentioned frequently, the methods for pyrite measurement/detection should be included somewhere in section 2.

**Author Comment: A description of pyrite assessment was added at the end of Section 2.4**

Sect 4.1.1, paragraph 1 and 2: Suspected SMTZ presence is mentioned for the first time in paragraph 1, but evidence for a potential SMTZ at Lake Magadi is not presented until the next paragraph. L 317-319 states where SMTZ is suspected, we expect increases in methane GDGT indices, but L 340-342 states high methane GDGT indices contributes to evidence for SMTZ presence. The way this is written has some circular reasoning. The latter (using biomarker and geochemical evidence to make a statement) is more logical than the former (using an assumption to suggest expected patterns in biomarker and geochemical evidence).

**Author Comment: Our logic of flow was to state the likely environment (SMTZ) which would result in our observations and then we explain how our results fit the environment and then conclude by stating that the combined evidence is likely pointing to it being an SMTZ environment. This seems logical to us.**

L 385: If meromixis is a present characteristic of Lake Magadi, should explicitly state this fact in sect 2.1 study location description.

**Author Comment: It is not a present characteristic of Lake Magadi, and the language has been clarified on L 381 by adding "in this study" to help define that these are not modern interpretations but paleo interpretations.**

Sect 4.1.2, paragraph 3: Both Ca/Na and REEs are mentioned here. It may merit briefly

reminding/explaining what each represent (e.g., REEs come from hydrothermal inputs) as neither was explicitly defined in the context of Lake Magadi prior to this section.

**Author Comment: Section 3.3 had the line "Increased values of REEs are characteristic of sodic systems influenced by hydrothermal springs, namely such as Mono Lake in California and this system (Johannesson and Lyons, 1994; Owen et al., 2019)." We have added an additional description of Ca/Na and why it is a useful metric for the core' interpretations on L 286.**

Sect 4.1.2, last paragraph: Delete "The $CPI_{alk}$ and $CPI_{FA}$ 510 averages were 4.6 and 5.0, indicating more terrestrial input." Consider also deleting the following sentence, "So, while these values are lower than Interval 3, and closer to the values in Interval 1, these still indicate a higher terrestrial input during this timeframe," as now there's no longer a discussion of terrestrial inputs. If this is deleted, then the only sentence left in the paragraph (starting with "Lastly, ...") would better be moved to the end of the previous paragraph to prevent having an orphan sentence.

**Author Comment: The reviewer's comments have been implemented in the manuscript.**

**Technical Corrections**

**Author Comment: All technical corrections were implemented and as for the L 586 comment about the manuscript number being "#XXX", yes this will be changed once it is finally accepted for publication. This may take some extra time as our collaborator (Andrew S Cohen) who archived these numbers has recently passed away. The issue with coloration on Fig. 2 has also been resolved.**

L 17-18: Replace "Ml "switches off" (Ml< 0.2); and on (Ml> 0.5)" with "The Ml switches "off" (Ml< 0.2) and "on" (Ml > 0.5)"

L 43: Change "generation of" to

"the"

L 69: Remove "Moreover"

L 110: Remove "More recently" as it is used to start the next

paragraph. L 120: Remove "Various"

L 121-122: This line, "Although Lake Magadi is situated near the equator, it lies in a rain shadow. Consequently, today it has a large moisture deficit (2400 mm evaporation versus 500 mm precipitation annually: Damnati and Taieb, 1995)" would better fit as the last line of sect 2.1, paragraph 1, rather than where it is currently at.

L 211: Change "was used here" to "is used here"

Equation 4: Un-bold the text

L 222: Change "intervals of focus (i.e., Intervals 1, 3, and 5)" to "intervals of focus in the sediment core (designated as intervals 1, 3, and 5 in the results and discussion sections)"

L 227-231: Condense these lines, starting from "Prior to performing...", to "The data was tested for normality via the built-in "Normality and Lognormality Tests" function in GraphPad©. Tests yielded lognormal distributions of each dataset and found the data to be non-normally distributed."

L 246: Is "197 to 149 ka" supposed to have a "ca." before it like the other

intervals? L 249: Remove "Table SlTable Sl"

L 262: Change "46.6815 m" to "46.68 m"

L 263: Remove spaces in "%GDGT-0 / cren"

L 277: Remove "Table Sl" from the start of the

sentences. L 291: Change "namely" to "such as"

L 344: This seems to refer to Figure 5 rather Figure 4.

L 345-346: This seems to refer to Figure 2 and 5 rather than 3 and 5.

L 346-347: Change "resulting from a more prevalent" to "as seen in a high".

L 354: Change "reported values" to "reported $6^{13}$CoMvalues" (assuming Summons et al. (1998) reported the $6^{13}$C for OM)

L 402: Put Table Sl in parenthesis.

L 406-407: Split up the sentence at "and while", ending the first with "of the Ml" and starting the next with "while both".

L 420: Flip "(Ml-on; Fig. 2)" to "(Fig. 2; Ml-on)" for consistency with the earlier parathesis.

L 424: Change "such as hot spring mats in Thermoproteota" to "such as in hot spring mats made by Thermoproteota".

L 446-449: Shorten the two sentences (as they're currently a bit redundant) to something like "In agreement with the pollen record, the $6^{13}$CoMvalues likely record a mixture of C4 grasses and C4 sedges. Similar $6^{13}$C values were reported in C27 to C33 n-alkanes in equatorial regions of Cameroon, ranging from
-18.2 to -17.6 %0 and recording the signals from C4 grasses and sedges (Garcin et al., 2014)."

L 497: Can delete "(averaging 1.5 excluding the outlying value of 36.7). This average is closer to shallow Group I.la Thermoproteota as described previously" as much of this was already stated in an

earlier sentence.

L 526-527: Add a comma between "Magadi ash" so it's "Magadi,

ash". L 545: Remove spaces in "Ca/ Na" so it becomes "Ca/Na".

L 551: This seems to refer to Fig. 4a rather than Sb.

L 558: Change "spring-runoff ratios" to "spring/runoff ratios" for consistency.

L 580: Should replace "RBO" with the full name of the co-author.

L 581: Change "special thank" to "special thanks".

L 582: Change "for providing, who provided" to "for

providing". L 584: Change "CSDF" to "CSD".

L 586: I assume "#XXX" will be replaced with an appropriate publication number once this paper is officially published?

Fig 2d: There seems to be a layer order issue with the data points in intervals 2, 4, and 6 plotting behind the yellow box instead of in front.

L 983-984: Remove the spaces between all the ratio indices (i.e., "%0/Cren" instead of "% 0/ Cren").

L 985-986: Remove "Checkered patterns indicate periods of tuffaceous silt deposit, which align with the low Ml intervals" as there is no longer a checkered pattern in the figure.

L 991: Add "and in the Y-axis scale for $6^{13}$CoM" after "X-axis scale".

L 995-996: Remove "The checkered pattern is indicative of periods of higher inferred hydrothermal flow" as there is no longer a checkered pattern in the figure.

**Referee #2**

I am satisfied with the changes made by the authors and would now support publication in Biogeosciences after a few final corrections:

**Author Comment: Each of these have been completed. Figure S1 has been updated with the relevant information.**

Abstract, l. 18: Please delete semicolon.

Title of section 2.2, l. 153: "Leaf wax" should be deleted (maybe replace by "GDGT")

Section 3.1, l. 249: Please check positioning of reference to table S1 (appears twice).

Section 4.1.1, l. 402: Please check positioning of reference to table S1.

Section 4.2, l. 551–554: References to Fig. 5a or b should be references to Fig 4a or b.

Fig. S1: Please label the main compounds in each chromatogram and add an explanation on the type of the chromatograms into the caption (e.g. TIC or ion filter?; LC chromatogram).

---

## Author Response (AR3)

Notification to the authors:

1) Please add the "†" after the author's name. In addition, please write "† deceased" below the last affiliation (7).

Author Comment: This has been corrected.

2) Please add doi numbers to the references in the reference list (where available).

Author Comment: Added all available DOI numbers.

3) Regarding Figure 1, please add the respective copyright statement or image credit if the aerial photo has neither been taken by you or your co-authors. Please also see https://www.biogeosciences.net/submission.html#mapsaerials

Author Comment: Added "Credit ESRI 2025".